# Learning Abstract Options

**Matthew Riemer, Miao Liu, and Gerald Tesauro**
IBM Research
T.J. Watson Research Center, Yorktown Heights, NY
{mdriemer, miao.liu1, gtesauro}@us.ibm.com

## Abstract

Building systems that autonomously create temporal abstractions from data is a
key challenge in scaling learning and planning in reinforcement learning. One
popular approach for addressing this challenge is the options framework [29].
However, only recently in [1] was a policy gradient theorem derived for online
learning of general purpose options in an end to end fashion. In this work, we
extend previous work on this topic that only focuses on learning a two-level
hierarchy including options and primitive actions to enable learning simultaneously
at multiple resolutions in time. We achieve this by considering an arbitrarily deep
hierarchy of options where high level temporally extended options are composed
of lower level options with finer resolutions in time. We extend results from
[1] and derive policy gradient theorems for a deep hierarchy of options. Our
proposed *hierarchical option-critic* architecture is capable of learning internal
policies, termination conditions, and hierarchical compositions over options without
the need for any intrinsic rewards or subgoals. Our empirical results in both discrete
and continuous environments demonstrate the efficiency of our framework.

## 1 Introduction

In reinforcement learning (RL), *options* [29, 21] provide a general framework for defining temporally
abstract courses of action for learning and planning. Discovering these temporal abstractions au-
tonomously has been the subject of extensive research [16, 28, 17, 27, 26] with approaches that can be
used in continuous state and/or action spaces only recently becoming feasible [9, 20, 15, 14, 10, 31, 3].
Most existing work has focused on finding subgoals (i.e. useful states for the agent to reach) and
then learning policies to achieve them. However, these approaches do not scale well because of their
combinatorial nature. Recent work on *option-critic* learning blurs the line between option discovery
and option learning by providing policy gradient theorems for optimizing a two-level hierarchy of op-
tions and primitive actions [1]. These approaches have achieved success when applied to Q-learning
on Atari games, but also in continuous action spaces [7] and with asynchronous parallelization [6].
In this paper, we extend option-critic to a novel *hierarchical option-critic* framework, presenting
generalized policy gradient theorems that can be applied to an arbitrarily deep hierarchy of options.

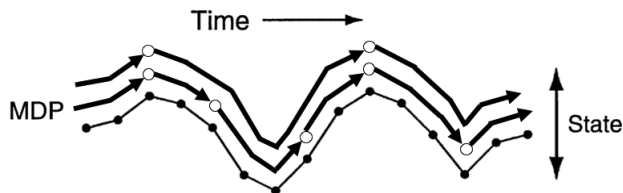

Figure 1: State trajectories over a three-level hierarchy of options. Open circles represent SMDP
decision points while filled circles are primitive steps within an option. The low level options are
temporally extended over primitive actions, and high level options are even further extended.

Work on learning with temporal abstraction is motivated by two key potential benefits over learning with primitive actions: long term credit assignment and exploration. Learning only at the primitive action level or even with low levels of abstraction slows down learning, because agents must learn longer sequences of actions to achieve the desired behavior. This frustrates the process of learning in environments with sparse rewards. In contrast, agents that learn a high level decomposition of sub-tasks are able to explore the environment more effectively by exploring in the abstract action space rather than the primitive action space. While the recently proposed deliberation cost [6] can be used as a margin that effectively controls how temporally extended options are, the standard two-level version of the option-critic framework is still ill-equipped to learn complex tasks that require sub-task decomposition at multiple quite different temporal resolutions of abstraction. In Figure 1 we depict how we overcome this obstacle to learn a deep hierarchy of options. The standard two-level option hierarchy constructs a Semi-Markov Decision Process (SMDP), where new options are chosen when temporally extended sequences of primitive actions are terminated. In our framework, we consider not just options and primitive actions, but also an arbitrarily deep hierarchy of lower level and higher level options. Higher level options represent a further temporally extended SMDP than the low level options below as they only have an opportunity to terminate when all lower level options terminate.

We will start by reviewing related research and by describing the seminal work we build upon in this paper that first derived policy gradient theorems for learning with primitive actions [30] and options [1]. We will then describe the core ideas of our approach, presenting hierarchical intra-option policy and termination gradient theorems. We leverage this new type of policy gradient learning to construct a *hierarchical option-critic* architecture, which generalizes the *option-critic* architecture to an arbitrarily deep hierarchy of options. Finally, we demonstrate the empirical benefit of this architecture over standard *option-critic* when applied to RL benchmarks. To the best of our knowledge, this is the first general purpose end-to-end approach for learning a deep hierarchy of options beyond two-levels in RL settings, scaling to very large domains at comparable efficiency.

## 2   Related Work

Our work is related to recent literature on learning to compose skills in RL. As an example, Sahni et al. [23] leverages a logic for combining pre-learned skills by learning an embedding to represent the combination of a skill and state. Unfortunately, their system relies on a pre-specified sub-task decomposition into skills. In [25], the authors propose to ground all goals in a natural language description space. Created descriptions can then be high level and express a sequence of goals. While these are interesting directions for further exploration, we will focus on a more general setting without provided natural language goal descriptions or sub-task decomposition information.

Our work is also related to methods that learn to decompose the problem over long time horizons. A prominent paradigm for this is Feudal Reinforcement Learning [4], which learns using manager and worker models. Theoretically, this can be extended to a deep hierarchy of managers and their managers as done in the original work for a hand designed decomposition of the state space. Much more recently, Vezhnevets et al. [32] showed the ability to successfully train a Feudal model end to end with deep neural networks for the Atari games. However, this has only been achieved for a two-level hierarchy (i.e. one manager and one worker). We can think of Feudal approaches as learning to decompose the problem with respect to the state space, while the options framework learns a temporal decomposition of the problem. Recent work [11] also breaks down the problem over a temporal hierarchy, but like [32] is based on learning a latent goal representation that modulates the policy behavior as opposed to options. Conceptually, options stress choosing among skill abstractions and Feudal approaches stress the achievement of certain kinds of states. Humans tend to use both of these kinds of reasoning when appropriate and we conjecture that a hybrid approach will likely win out in the end. Unfortunately, in the space available we feel that we cannot come to definitive conclusions about the precise nature of the differences and potential synergies of these approaches.

The concept of learning a hierarchy of options is not new. It is an obviously desirable extension of options envisioned in the original papers. However, actually learning a deep hierarchy of options end to end has received surprisingly little attention to date. Compositional planning where options select other options was first considered in [26]. The authors provided a generalization of value iteration to option models for multiple subgoals, leveraging explicit subgoals for options. Recently, Fox et al. [5] successfully trained a hierarchy of options end to end for imitation learning. Their approach leverages an EM based algorithm for recursive discovery of additional levels of the option hierarchy.

Unfortunately, their approach is only applicable to imitation learning and not general purpose RL. We are the first to propose theorems along with a practical algorithm and architecture to train arbitrarily deep hierarchies of options end to end using policy gradients, maximizing the expected return.

## 3    Problem Setting and Notation

A Markov Decision Process (MDP) consists of a set of states $\mathcal{S}$, a set of actions $\mathcal{A}$, a transition function $\mathcal{P} : \mathcal{S} \times \mathcal{A} \to (\mathcal{S} \to [0,1])$ and a reward function $r : \mathcal{S} \times \mathcal{A} \to \mathbb{R}$. We follow [1] and develop our ideas assuming discrete state and action sets, while our results extend to continuous spaces using usual measure-theoretic assumptions as we demonstrate in our experiments. A policy is a probability distribution over actions conditioned on states, $\pi : \mathcal{S} \to \mathcal{A} \to [0,1]$. The value function of a policy $\pi$ is defined as the expected return $V_\pi(s) = \mathbb{E}_\pi[\sum_{t=0}^\infty \gamma^t r_{t+1}|s_0 = s]$ with an action-value function of $Q_\pi(s,a) = \mathbb{E}_\pi[\sum_{t=0}^\infty \gamma^t r_{t+1}|s_0 = s, a_0 = a]$ where $\gamma \in [0,1)$ is the *discount factor*.

**Policy gradient methods** [30, 8] address the problem of finding a good policy by performing stochastic gradient descent to optimize a performance objective over a given family of parametrized stochastic policies, $\pi_\theta$. The policy gradient theorem [30] provides an expression for the gradient of the discounted reward objective with respect to $\theta$. The objective is defined with respect to a designated start state (or distribution) $s_0 : \rho(\theta, s_0) = \mathbb{E}_{\pi_\theta}[\sum_{t=0}^\infty \gamma^t r_{t+1}|s_0]$. The policy gradient theorem shows that: $\frac{\partial \rho(\theta, s_0)}{\partial \theta} = \sum_s \mu_{\pi_\theta}(s|s_0) \sum_a \frac{\partial \pi_\theta(a|s)}{\partial \theta} Q_{\pi_\theta}(s,a)$, where $\mu_{\pi_\theta}(s|s_0) = \sum_{t=0}^\infty \gamma^t P(s_t = s|s_0)$ is a discounted weighting of the states along the trajectories starting from $s_0$.

**The options framework** [29, 21] formalizes the idea of temporally extended actions. A Markovian option $o \in \Omega$ is a triple $(I_o, \pi_o, \beta_o)$ in which $I_o \subseteq S$ is an initiation set, $\pi_o$ is an intra-option policy, and $\beta_o : \mathcal{S} \to [0,1]$ is a termination function. Like most option discovery algorithms, we assume that all options are available everywhere. MDPs with options become SMDPs [22] with a corresponding optimal value function over options $V_\Omega^*(s)$ and option-value function $Q_\Omega^*(s,o)$ [29, 21].

**The option-critic architecture** [1] leverages a call-and-return option execution model, in which an agent picks option $o$ according to its policy over options $\pi_\Omega(o|s)$, then follows the intra-option policy $\pi(a|s,o)$ until termination (as dictated by $\beta(s,o)$), at which point this procedure is repeated. Let $\pi_\theta(a|s,o)$ denote the intra-option policy of option $o$ parametrized by $\theta$ and $\beta_\phi(s,o)$ the termination function of $o$ parameterized by $\phi$. Like policy gradient methods, the option-critic architecture optimizes directly for the discounted return expected over trajectories starting at a designated state $s_0$ and option $o_0$: $\rho(\Omega, \theta, \phi, s_0, o_0) = \mathbb{E}_{\Omega, \pi_\theta, \beta_\phi}[\sum_{t=0}^\infty \gamma^t r_{t+1}|s_0, o_0]$. The option-value function is then:

$$Q_\Omega(s,o) = \sum_a \pi_\theta(a|s,o) Q_U(s,o,a), \qquad (1)$$

where $Q_U : \mathcal{S} \times \Omega \times \mathcal{A} \to \mathbb{R}$ is the value of executing an action in the context of a state-option pair:

$$Q_U(s,o,a) = r(s,a) + \gamma \sum_{s'} P(s'|s,a) U(s',o). \qquad (2)$$

The $(s,o)$ pairs lead to an augmented state space [12]. The option-critic architecture instead leverages the function $U : \Omega \times \mathcal{S} \to \mathbb{R}$ which is called the option-value function upon arrival [29]. The value of executing $o$ upon entering state $s'$ is given by:

$$U(s',o) = (1 - \beta_\phi(s',o)) Q_\Omega(s',o) + \beta_\phi(s',o) V_\Omega(s'). \qquad (3)$$

$Q_U$ and $U$ both depend on $\theta$ and $\phi$, but are omitted from the notation for clarity. The intra-option policy gradient theorem results from taking the derivative of the expected discounted return with respect to the intra-option policy parameters $\theta$ and defines the update rule for the intra-option policy:

$$\frac{\partial Q_\Omega(s_0, o_0)}{\partial \theta} = \sum_{s,o} \mu_\Omega(s,o|s_0,o_0) \sum_a \frac{\partial \pi_\theta(a|s,o)}{\partial \theta} Q_U(s,o,a). \qquad (4)$$

where $\mu_\Omega(s,o|s_0,o_0)$ is a discounted weighting of state-option pairs along trajectories starting from $(s_0, o_0) : \mu_\Omega(s,o|s_0,o_0) = \sum_{t=0}^\infty \gamma^t P(s_t = s, o_t = o|s_0, o_0)$. The termination gradient theorem results from taking the derivative of the expected discounted return with respect to the termination policy parameters $\phi$ and defines the update rule for the termination policy for the initial condition $(s_1, o_0)$:

$$\frac{\partial Q_\Omega(s,o)}{\partial \phi} = \sum_a \pi_\theta(a|s,o) \sum_{s'} \gamma P(s'|s,a) \frac{\partial U(s',o)}{\partial \phi}, \qquad (5)$$

$$\frac{\partial U(s_1, o_0)}{\partial \phi} = -\sum_{s',o} \mu_\Omega(s', o | s_1, o_o) \frac{\partial \beta_\phi(s', o)}{\partial \phi} A_\Omega(s', o), \tag{6}$$

where $\mu_\Omega$ is a discounted weighting of $(s, o)$ from $(s_1, o_0)$ : $\mu_\Omega(s, o | s_1, o_0) = \sum_{t=0}^{\infty} \gamma^t P(s_{t+1} = s, o_t = o | s_1, o_0)$. $A_\Omega$ is the advantage function over options: $A_\Omega(s', o) = Q_\Omega(s', o) - V_\Omega(s')$.

# 4 Learning Options with Arbitrary Levels of Abstraction

**Notation:** As it makes our equations much clearer and more condensed we adopt the notation $x^{i:i+j} = x^i, ..., x^{i+j}$. This implies that $x^{i:i+j}$ denotes a list of variables in the range of $i$ through $i+j$.

**The hierarchical options framework** that we introduce in this work considers an agent that learns using an $N$ level hierarchy of policies, termination functions, and value functions. Our goal is to extend the ideas of the option-critic architecture in such a way that our framework simplifies to policy gradient based learning when $N = 1$ and option-critic learning when $N = 2$. At each hierarchical level above the lowest primitive action level policy, we consider an available set of options $\Omega^{1:N-1}$ that is a subset of the total set of available options $\Omega$. This way we keep our view of the possible available options at each level very broad. On one extreme, each hierarchical level may get its own unique set of options and on the other extreme each hierarchical level may share the same set of options. We present a diagram of our proposed architecture in Figure 2.

We denote $\pi^1_{\theta^1}(o^1 | s)$ as the policy over the most abstract options in the hierarchy $o^1 \in \Omega_1$ given the state $s$. For example, $\pi^1 = \pi_\Omega$ from our discussion of the option-critic architecture. Once $o^1$ is chosen with policy $\pi^1$, then we go to policy $\pi^2_{\theta^2}(o^2 | s, o^1)$, which is the next highest level policy, to select $o^2 \in \Omega_2$ conditioning it on both the current state $s$ and the selected highest level option $o^1$. This process continues on in the same fashion stepping down to policies at lower levels of abstraction conditioned on the augmented state space considering

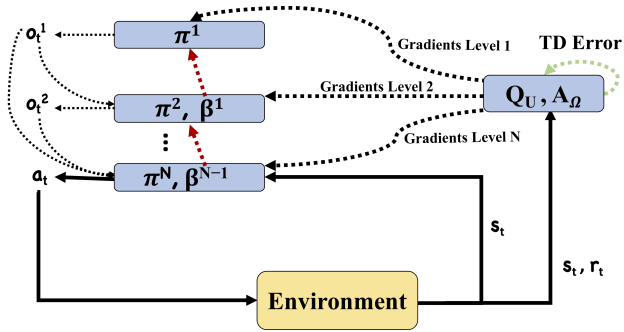

Figure 2: A diagram describing our proposed hierarchical option-critic architecture. Dotted lines represent processes within the agent while solid lines represent processes within the environment. Option selection is top down through the hierarchy and option termination is bottom up (represented with red dotted lines).

all selected higher level options until we reach policy $\pi^N_{\theta^N}(a | s, o^{1:N-1})$. $\pi^N$ is the lowest level policy and it finally selects over the primitive action space conditioned on all of the selected options.

Each level of the option hierarchy has a complimentary termination function $\beta^1_{\phi^1}(s, o^1), ..., \beta^{N-1}_{\phi^{N-1}}(s, o^{1:N-1})$ that governs the termination pattern of the selected option at that level. We adopt a bottom up termination strategy where high level options only have an opportunity to terminate when all of the lower level options have terminated first. For example, $o^{N-2}_t$ cannot terminate until $o^{N-1}_t$ terminates at which point we can assess $\beta^{N-2}_{\phi^{N-2}}(s, o^{1:N-2})$ to see whether $o^{N-2}_t$ terminates. If it did terminate, this would allow $o^{N-3}_t$ the opportunity to asses if it should terminate and so on. This condition ensures that higher level options will be more temporally extended than their lower level option building blocks, which is a key motivation of this work.

The final key component of our system is the value function over the augmented state space. To enable comprehensive reasoning about the policies at each level of the option hierarchy, we need to maintain value functions that consider the state and every possible combination of active options and actions $V_\Omega(s), Q_\Omega(s, o^1), ..., Q_\Omega(s, o^{1:N-1}, a)$. These value functions collectively serve as the *critic* in our analogy to the actor-critic and option-critic training paradigms.

## 4.1 Generalizing the Option Value Function to N Hierarchical Levels

Like policy gradient methods and option-critic, the hierarchical options framework optimizes directly for the discounted return expected over all trajectories starting at a state $s_0$ with active options $o_0^{1:N-1}$:

$$\rho(\Omega^{1:N-1}, \theta^{1:N}, \phi^{1:N-1}, s_0, o_0^{1:N-1}) = \mathbb{E}_{\Omega^{1:N-1}, \pi_\theta^{1:N}, \beta_\phi^{1:N-1}}\left[\sum_{t=0}^\infty \gamma^t r_{t+1} | s_0, o_0^{1:N-1}\right] \tag{7}$$

This return depends on the policies and termination functions at each level of abstraction. We now consider the option value function for understanding reasoning about an option $o^\ell$ at level $1 \leq \ell \leq N$ based on the augmented state space $(s, o^{1:\ell-1})$:

$$Q_\Omega(s, o^{1:\ell-1}) = \sum_{o^\ell} \pi_{\theta^\ell}^\ell(o^\ell | s, o^{1:\ell-1}) Q_U(s, o^{1:\ell}) \tag{8}$$

Note that in order to simplify our notation we write $o^\ell$ as referring to both abstract and primitive actions. As a result, $o^N$ is equivalent to $a$, leveraging the primitive action space $\mathcal{A}$. Extending the meaning of $Q_U$ from [1], we define the corresponding value of executing an option in the presence of the currently active higher level options by integrating out the lower level options:

$$Q_U(s, o^{1:\ell}) = \sum_{o^N}\cdots\sum_{o^{\ell+1}} \prod_{j=\ell+1}^N \pi^j(o^j | s, o^{1:j-1})[r(s, o^N) + \gamma\sum_{s'} P(s' | s, o^N) U(s', o^{1:\ell-1})]. \tag{9}$$

The hierarchical option value function upon arrival $U$ with augmented state $(s, o^{1:\ell-1})$ is defined as:

$$U(s', o^{1:\ell-1}) = \underbrace{(1 - \beta_{\phi^{N-1}}^{N-1}(s', o^{1:N-1}))Q_\Omega(s', o^{1:\ell-1})}_{\text{none terminate } (N \geq 1)} + \underbrace{V_\Omega(s') \prod_{j=N-1}^1 \beta_{\phi^j}^j(s', o^{1:j})}_{\text{all options terminate } (N \geq 2)} +$$

$$\underbrace{Q_\Omega(s', o^{1:\ell-1}) \sum_{q=N-1}^\ell (1 - \beta_{\phi^{q-1}}^{q-1}(s', o^{1:q-1})) \prod_{z=N-1}^q \beta_{\phi^z}^z(s', o^{1:z})}_{\text{only lower level options terminate } (N \geq 3)} + \tag{10}$$

$$\underbrace{\sum_{i=1}^{\ell-2} (1 - \beta_{\phi^i}^i(s', o^{1:i}))Q_\Omega(s', o^{1:i}) \prod_{k=i+1}^{N-1} \beta_{\phi^k}^k(s', o^{1:k})]}_{\text{some relevant higher level options terminate } (N \geq 3)}.$$

We explain the derivation of this equation [1] in the Appendix 1.1. Finally, before we can extend the policy gradient theorem, we must establish the Markov chain along which we can measure performance for options with $N$ levels of abstraction. This is derived in the Appendix 1.2.

## 4.2 Generalizing the Intra-option Policy Gradient Theorem

We can think of actor-critic architectures, generalizing to the option-critic architecture as well, as pairing a critic with each actor network so that the critic has additional information about the value of the actor's actions that can be used to improve the actor's learning. However, this is derived by taking gradients with respect to the parameters of the policy while optimizing for the expected discounted return. The discounted return is approximated by a critic (i.e. value function) with the same augmented state-space as the policy being optimized for. As examples, an actor-critic policy $\pi(a|s)$ is optimized by taking the derivative of its parameters with respect to $V_\pi(s)$ [30] and an option-critic policy $\pi(a|s, o)$ is optimized by taking the derivative of its parameters with respect to $Q_\Omega(s, o)$ [1]. The intra-option policy gradient theorem [1] is an important contribution, outlining how to optimize for a policy that is also associated with a termination function. As the policy over options in that work never terminates, it does not need a special training methodology and the option-critic architecture allows the practitioner to pick their own method of learning the policy over options while using Q Learning as an example in their experiments. We do the same for our highest level policy $\pi^1$ that also never terminates. For all other policies $\pi^{2:N}$ we perform a generalization of actor-critic learning by providing a critic at each level and guiding gradients using the appropriate critic.

We now seek to generalize the intra-option policy gradients theorem, deriving the update rule for a policy at an arbitrary level of abstraction $\pi^\ell$ by taking the gradient with respect to $\theta^\ell$ using the value function with the same augmented state space $Q_\Omega(s, o^{1:\ell-1})$. Substituting from equation (8) we find:

$$\frac{\partial Q_\Omega(s, o^{1:\ell-1})}{\partial \theta^\ell} = \frac{\partial}{\partial \theta^\ell} \sum_{o^\ell} \pi_{\theta^\ell}^\ell(o^\ell|s, o^{1:\ell-1}) Q_U(s, o^{1:\ell}). \tag{11}$$

**Theorem 1** (Hierarchical Intra-option Policy Gradient Theorem). *Given an N level hierarchical set of Markov options with stochastic intra-option policies differentiable in their parameters $\theta^\ell$ governing each policy $\pi^\ell$, the gradient of the expected discounted return with respect to $\theta^\ell$ and initial conditions $(s_0, o_0^{1:N-1})$ is:*

$$\sum_{s, o^{1:\ell-1}} \mu_\Omega(s, o^{1:\ell-1}|s_0, o_0^{1:N-1}) \sum_{o^\ell} \frac{\partial \pi_{\theta^\ell}^\ell(o^\ell|s, o^{1:\ell-1})}{\partial \theta^\ell} Q_U(s, o^{1:\ell}),$$

where $\mu_\Omega$ is a discounted weighting of augmented state tuples along trajectories starting from $(s_0, o_0^{1:N-1}) : \mu_\Omega(s, o^{1:\ell-1}|s_0, o_0^{1:N-1}) = \sum_{t=0}^\infty \gamma^t P(s_t = s, o_t^{1:\ell-1} = o^{1:\ell-1}|s_0, o_0^{1:N-1})$. A proof is in Appendix 1.3.

### 4.3 Generalizing the Termination Gradient Theorem

We now turn our attention to computing gradients for the termination functions $\beta^\ell$ at each level, assumed to be stochastic and differentiable with respect to the associated parameters $\phi^\ell$.

$$\frac{\partial Q_\Omega(s, o^{1:\ell})}{\partial \phi^\ell} = \sum_{o^N} \cdots \sum_{o^{\ell+1}} \prod_{j=\ell+1}^N \pi^j(o^j|s, o^{1:j-1}) \gamma \sum_{s'} P(s'|s, o^N) \frac{\partial U(s', o^{1:\ell})}{\partial \phi^\ell} \tag{12}$$

Hence, the key quantity is the gradient of $U$. This is a natural consequence of call-and-return execution, where termination function quality can only be evaluated upon entering the next state.

**Theorem 2** (Hierarchical Termination Gradient Theorem). *Given an N level hierarchical set of Markov options with stochastic termination functions differentiable in their parameters $\phi^\ell$ governing each function $\beta^\ell$, the gradient of the expected discounted return with respect to $\phi^\ell$ and initial conditions $(s_1, o_0^{1:N-1})$ is:*

$$- \sum_{s, o^{1:\ell}} \mu_\Omega(s, o^{1:\ell}|s_1, o_0^{1:N-1}) \prod_{i=\ell+1}^{N-1} \beta_{\phi^i}^i(s, o^{1:i}) \frac{\partial \beta_{\phi^\ell}^\ell(s, o^{1:\ell})}{\partial \phi^\ell} A_\Omega(s, o^{1:\ell})$$

where $\mu_\Omega$ is a discounted weighting of augmented state tuples along trajectories starting from $(s_1, o_0^{1:N-1}) : \mu_\Omega(s, o^{1:\ell-1}|s_1, o_0^{1:N-1}) = \sum_{t=0}^\infty \gamma^t P(s_t = s, o_t^{1:\ell} = o^{1:\ell}|s_1, o_0^{1:N-1})$. $A_\Omega$ is the generalized advantage function over a hierarchical set of options $A_\Omega(s', o^{1:\ell}) = Q_\Omega(s', o^{1:\ell}) - V_\Omega(s)[\prod_{j=\ell-1}^1 \beta_{\phi^j}^j(s', o^{1:j})] - \sum_{i=1}^{\ell-1}(1 - \beta_{\phi^i}^i(s', o^{1:i})) Q_\Omega(s', o^{1:i})[\prod_{k=i+1}^{\ell-1} \beta_{\phi^k}^k(s', o^{1:k})]$. $A_\Omega$ compares the advantage of not terminating the current option with a probability weighted expectation based on the likelihood that higher level options also terminate. In [1] this expression was simple as there was not a hierarchy of higher level termination functions to consider. A proof is in Appendix 1.4.

It is interesting to see the emergence of an advantage function as a natural consequence of the derivation. As in [1] where this kind of relationship also appears, the advantage function gives the theorem an intuitive interpretation. When the option choice is sub-optimal at level $\ell$ with respect to the expected value of terminating option $\ell$, the advantage function is negative and increases the odds of terminating that option. A new concept, not paralleled in the option-critic derivation, is the inclusion of a $\prod_{i=\ell+1}^{N-1} \beta_{\phi^i}^i(s, o^{1:i})$ multiplicative factor. This can be interpreted as discounting gradients by the likelihood of this termination function being assessed as $\beta^\ell$ is only used if all lower level options terminate. This is a natural consequence of multi-level call and return execution.

## 5 Experiments

We would now like to empirically validate the efficacy of our proposed hierarchical option-critic (HOC) model. We achieve this by exploring benchmarks in the tabular and non-linear function approximation settings. In each case we implement an agent that is restricted to primitive actions (i.e. $N = 1$), an agent that leverages the option-critic (OC) architecture (i.e. $N = 2$), and an agent with the HOC architecture at level of abstraction $N = 3$. We will demonstrate that complex RL problems may be more easily learned using beyond two levels of abstraction and that the HOC architecture can successfully facilitate this level of learning using data from scratch.

For our tabular architectures, we followed protocol from [1] and chose to parametrize the intra-option policies with softmax distributions and the terminations with sigmoid functions. The policy over options was learned using intra-option Q-learning. We also implemented primitive actor-critic (AC) using a softmax policy. For the nonlinear function approximation setting, we trained our agents using A3C [19]. Our primitive action agents conduct A3C training using a convolutional network when there is image input followed by an LSTM

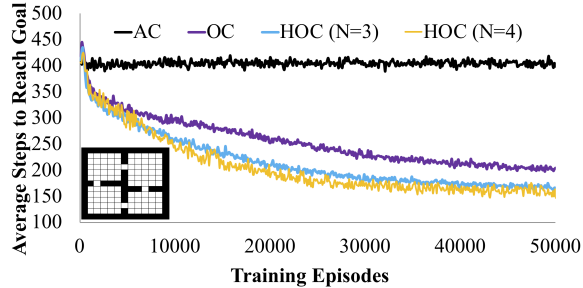

Figure 3: Learning performance as a function of the abstraction level for a nonstationary four rooms domain where the goal location changes every episode.

to contextualize the state. This way we ensure that benefits seen from options are orthogonal to those seen from these common neural network building blocks. We follow [6] to extend A3C to the Asynchronous Advantage Option-Critic (A2OC) and Asynchronous Advantage Hierarchical Option-Critic architectures (A2HOC). We include detailed algorithm descriptions for all of our experiments in Appendix 2. We also conducted hyperparameter optimization that is summarized along with detail on experimental protocol in Appendix 2. In all of our experiments, we made sure that the two-level OC architecture had access to more total options than the three level alternative and that the three level architecture did not include any additional hyperparameters. This ensures that empirical gains are the result of increasingly abstract options.

## 5.1 Tabular Learning Challenge Problems

**Exploring four rooms:** We first consider a navigation task in the four-rooms domain [29]. Our goal is to evaluate the ability of a set of options learned fully autonomously to learn an efficient exploration policy within the environment. The initial state and the goal state are drawn uniformly from all open non-wall cells every episode. This setting is highly non-stationary, since the goal changes every episode. Primitive movements can fail with probability $\frac{1}{3}$, in which case the agent transitions randomly to one of the empty adjacent cells. The reward is +1 at the goal and 0 otherwise. In Figure 3 we report the average number of steps taken in the last 100 episodes every 100 episodes, reporting the average of 50 runs with different random seeds for each algorithm. We can clearly see that reasoning with higher levels of abstraction is critical to achieving a good exploration

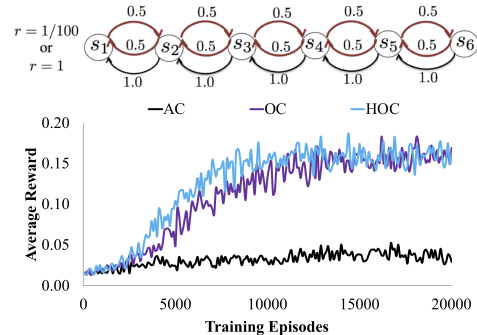

Figure 4: The diagram, from [10], details the stochastic decision process challenge problem. The chart compares learning performance across abstract reasoning levels.

policy and that reasoning with three levels of abstraction results in better sample efficient learning than reasoning with two levels of abstraction. For this experiment we explore four levels of abstraction as well, but unfortunately there seem to be diminishing returns at least for this tabular setting.

**Discrete stochastic decision process:** Next, we consider a hierarchical RL challenge problem as explored in [10] with a stochastic decision process where the reward depends on the history of visited states in addition to the current state. There are 6 possible states and the agent always starts at $s_2$. The agent moves left deterministically when it chooses left action; but the action right only succeeds half of the time, resulting in a left move otherwise. The terminal state is $s_1$ and the agent receives a reward of 1 when it first visits $s_6$ and then $s_1$. The reward for going to $s_1$ without visiting $s_6$ is 0.01. In Figure 4 we report the average reward over the last 100 episodes every 100 episodes, considering 10 runs with different random seeds for each algorithm. Reasoning with higher levels of abstraction is again critical to performing well at this task with reasonable sample efficiency. Both OC learning and HOC learning converge to a high quality solution surpassing performance obtained in [10]. However, it seems that learning converges faster with three levels of abstractions than it does with just two.

## 5.2 Deep Function Approximation Problems

**Multistory building navigation:** For an intuitive look at higher level reasoning, we consider the four rooms problem in a partially observed setting with an 11x17 grid at each level of a seven level building. The agent has a receptive field size of 3 in both directions, so observations for the agent are 9-dimension feature vectors with 0 in empty spots, 1 where there is a wall, 0.25 if there are stairs, or 0.5 if there is a goal location. The stairwells in the north east corner of the floor lead upstairs to the south west corner of the next floor up. Stairs in the south west corner of the floor lead down to the north east corner of the floor below. Agents start in a random location in the basement (which has no south west stairwell) and

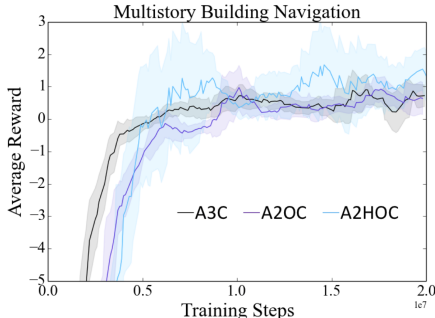

Figure 5: Building navigation learning performance across abstract reasoning levels.

must navigate to the roof (which has no north east stairwell) to find the goal in a random location. The reward is +10 for finding the goal and -0.1 for hitting a wall. This task could seemingly benefit from abstraction such as a composition of sub-policies to get to the stairs at each intermediate level. We report the rolling mean and standard deviation of the reward. In Figure 5 we see a qualitative difference between the policies learned with three levels of abstraction which has high variance, but fairly often finds the goal location and those learned with less abstraction. A2OC and A3C are hovering around zero reward, which is equivalent to just learning a policy that does not run into walls.

**Learning many Atari games with one model:** We finally consider application of the HOC to the Atari games [2]. Evaluation protocols for the Atari games are famously inconsistent [13], so to ensure for fair comparisons we implement apples to apples versions of our baseline architectures deployed with the same code-base and environment settings. We put our models to the test and consider a very challenging setting [24] where a single agent attempts to learn many Atari games at the same time. Our agents at-

| Architecture | Clipped Reward |
|---|---|
| A3C | 8.43 $_{\pm 2.29}$ |
| A2OC | 10.56 $_{\pm 0.49}$ |
| A2HOC | **13.12** $_{\pm 1.46}$ |

Table 1: Average clipped reward per episode over 5 runs on 21 Atari games.

tempt to learn 21 Atari games simultaneously, matching the largest previous multi-task setting on Atari [24]. Our tasks are hand picked to fall into three categories of related games each with 7 games represented. The first category is games that include maze style navigation (e.g. MsPacman), the second category is mostly fully observable shooter games (e.g. SpaceInvaders), and the final category is partially observable shooter games (e.g. BattleZone). We train each agent by always sampling the game with the least training frames after each episode, ensuring the games are sampled very evenly throughout training. We also clip rewards to allow for equal learning rates across tasks [18]. We train each game for 10 million frames (210 million total) and report statistics on the clipped reward achieved by each agent when evaluating the policy without learning for another 3 million frames on each game across 5 separate training runs. As our main metric, we report the summary of how each multi-task agent maximizes its reward in Table 1. While all agents struggle in this difficult setting, HOC is better able to exploit commonalities across games using fewer parameters and policies.

**Analysis of Learned Options:** An advantage of the multi-task setting is it allows for a degree of quantitative interpretability regarding when and how options are used. We report characteristics of the agents with median performance during the evaluation period. A2OC with 16 options uses 5 options the bulk of the time with the rest of the time largely split among another 6 options (Figure 6). The average number of time steps between switching options has a pretty narrow range across games falling between 3.4 (Solaris) and 5.5 (MsPacman). In contrast, A2HOC with three options at each branch of the hierarchy learns to switch options at a rich range of temporal resolutions depending on the game. The high level options vary between an average of 3.2 (BeamRider) and 9.7 (Tutankham) steps before switching. Meanwhile, the low level options vary between an average of 1.5 (Alien) and 7.8 (Tutankham) steps before switching. In Appendix 2.4 we provide additional details about the average duration before switching options for each game. In Figure 7 we can see that the most used options for HOC are distributed pretty evenly across a number of games, while OC tends to specialize its options on a smaller number of games. In fact, the average share of usage dominated by a single game for the top 7 most used options is 40.9% for OC and only 14.7% for HOC. Additionally, we can

| Option | Usage |
|---|---|
| o=0 | 3.5% |
| o=1 | 0.7% |
| o=2 | 7.5% |
| o=3 | 10.6% |
| o=4 | 8.8% |
| o=5 | 31.2% |
| o=6 | 0.7% |
| o=7 | 0.7% |
| o=8 | 2.7% |
| o=9 | 0.7% |
| o=10 | 2.3% |
| o=11 | 23.0% |
| o=12 | 1.1% |
| o=13 | 2.6% |
| o=14 | 3.3% |
| o=15 | 0.7% |

| Environment | o=0 | o=2 | o=3 | o=4 | o=5 | o=8 | o=11 | o=13 | o=14 |
|---|---|---|---|---|---|---|---|---|---|
| Alien | 0.9% | 0.6% | 0.3% | 0.7% | 0.1% | 1.1% | 18.5% | 1.1% | 0.9% |
| Amidar | 0.8% | 0.5% | 0.3% | 0.5% | 0.1% | 1.1% | 18.6% | 1.1% | 0.9% |
| Assault | 0.9% | 5.0% | 0.3% | 1.5% | 7.3% | 1.4% | 0.1% | 56.9% | 1.0% |
| Atlantis | 77.0% | 0.4% | 0.3% | 0.3% | 0.1% | 1.1% | 0.1% | 1.1% | 0.9% |
| BankHeist | 0.9% | 0.6% | 0.3% | 0.5% | 0.1% | 1.1% | 18.6% | 1.1% | 0.9% |
| BattleZone | 0.9% | 0.4% | 37.3% | 1.1% | 0.1% | 1.1% | 0.1% | 1.2% | 7.4% |
| BeamRider | 0.9% | 1.4% | 0.4% | 0.3% | 13.6% | 1.1% | 0.1% | 1.1% | 0.9% |
| Berzerk | 1.0% | 0.6% | 0.3% | 0.3% | 13.6% | 1.1% | 0.1% | 1.1% | 0.9% |
| Carnival | 0.9% | 1.0% | 0.3% | 0.3% | 13.4% | 1.1% | 0.1% | 4.1% | 0.9% |
| Centipede | 1.2% | 2.7% | 3.3% | 0.6% | 10.3% | 2.0% | 0.2% | 16.3% | 2.3% |
| ChopperCommand | 0.9% | 0.5% | 0.3% | 0.5% | 0.1% | 1.1% | 18.6% | 1.1% | 0.9% |
| DemonAttack | 0.9% | 0.4% | 0.3% | 0.3% | 13.7% | 1.3% | 0.2% | 1.6% | 1.3% |
| Jamesbond | 0.9% | 0.4% | 0.3% | 0.3% | 13.2% | 4.4% | 0.1% | 1.1% | 0.9% |
| MsPacman | 0.9% | 0.5% | 0.3% | 49.1% | 0.1% | 1.1% | 0.1% | 1.1% | 0.9% |
| Phoenix | 0.9% | 49.2% | 1.4% | 3.6% | 0.1% | 1.1% | 1.1% | 1.1% | 0.9% |
| Riverraid | 0.9% | 0.7% | 0.3% | 5.1% | 0.1% | 1.1% | 16.8% | 1.1% | 0.9% |
| Solaris | 5.6% | 0.5% | 15.6% | 1.1% | 0.1% | 73.4% | 0.2% | 1.2% | 4.1% |
| SpaceInvaders | 1.3% | 27.3% | 10.4% | 0.3% | 0.2% | 1.2% | 0.1% | 2.9% | 33.7% |
| Tutankham | 1.0% | 2.7% | 27.5% | 0.4% | 0.1% | 1.1% | 0.2% | 1.1% | 37.5% |
| WizardOfWor | 0.9% | 3.2% | 0.3% | 32.3% | 0.1% | 1.1% | 5.6% | 1.2% | 0.9% |
| Zaxxon | 0.9% | 1.5% | 0.3% | 0.4% | 13.5% | 1.1% | 0.2% | 1.1% | 0.9% |

Figure 6: Option usage (left) and specialization across Atari games for the top 9 most used options (right) of a 16 option Option-Critic architecture trained in the many task learning setting.

| Options | Usage |
|---|---|
| $o^1$=0 | 27.9% |
| $o^1$=1 | 29.0% |
| $o^1$=2 | 43.2% |
| ($o^1$=0,$o^2$=0) | 23.8% |
| ($o^1$=0,$o^2$=1) | 4.0% |
| ($o^1$=0,$o^2$=2) | 0.0% |
| ($o^1$=1,$o^2$=0) | 9.7% |
| ($o^1$=1,$o^2$=1) | 11.1% |
| ($o^1$=1,$o^2$=2) | 8.3% |
| ($o^1$=2,$o^2$=0) | 10.1% |
| ($o^1$=2,$o^2$=1) | 14.6% |
| ($o^1$=2,$o^2$=2) | 18.5% |

| Environment | ($o^1$=0,$o^2$=0) | ($o^1$=0,$o^2$=1) | ($o^1$=0,$o^2$=2) | ($o^1$=1,$o^2$=0) | ($o^1$=1,$o^2$=1) | ($o^1$=1,$o^2$=2) | ($o^1$=2,$o^2$=0) | ($o^1$=2,$o^2$=1) | ($o^1$=2,$o^2$=2) |
|---|---|---|---|---|---|---|---|---|---|
| Alien | 0.7% | 0.0% | 0.0% | 0.4% | 0.7% | 0.6% | 14.8% | 11.4% | 6.9% |
| Amidar | 1.6% | 3.6% | 0.0% | 0.7% | 0.8% | 0.8% | 10.5% | 15.4% | 3.7% |
| Assault | 4.3% | 0.8% | 0.0% | 5.5% | 7.4% | 7.1% | 2.7% | 2.4% | 6.2% |
| Atlantis | 0.6% | 0.0% | 0.0% | 9.8% | 10.8% | 8.9% | 3.9% | 1.8% | 5.7% |
| BankHeist | 4.8% | 81.4% | 0.0% | 0.3% | 0.8% | 0.5% | 0.5% | 0.7% | 0.1% |
| BattleZone | 4.4% | 1.0% | 0.0% | 1.2% | 0.7% | 0.6% | 8.8% | 11.0% | 5.1% |
| BeamRider | 0.7% | 0.0% | 0.0% | 7.8% | 9.6% | 9.7% | 2.0% | 1.8% | 8.2% |
| Berzerk | 4.4% | 0.1% | 0.0% | 10.4% | 11.8% | 10.4% | 2.2% | 1.2% | 0.7% |
| Carnival | 5.6% | 3.4% | 0.0% | 6.5% | 5.8% | 7.8% | 1.9% | 2.8% | 4.1% |
| Centipede | 1.0% | 0.3% | 0.1% | 2.6% | 2.2% | 3.7% | 1.7% | 2.0% | 17.5% |
| ChopperCommand | 1.2% | 0.0% | 0.0% | 4.0% | 6.4% | 5.6% | 9.1% | 9.2% | 3.5% |
| DemonAttack | 1.8% | 0.3% | 38.7% | 14.8% | 9.5% | 11.1% | 2.0% | 1.8% | 2.4% |
| Jamesbond | 2.7% | 4.2% | 0.0% | 1.0% | 1.4% | 1.0% | 9.8% | 12.8% | 4.1% |
| MsPacman | 17.9% | 1.3% | 0.0% | 0.4% | 0.8% | 0.8% | 0.6% | 0.3% | 0.7% |
| Phoenix | 4.3% | 0.3% | 7.3% | 13.5% | 8.1% | 8.2% | 2.3% | 1.9% | 2.0% |
| Riverraid | 0.7% | 0.0% | 0.0% | 0.5% | 0.6% | 0.6% | 14.7% | 11.8% | 6.6% |
| Solaris | 9.8% | 0.4% | 0.0% | 7.8% | 6.0% | 7.9% | 0.9% | 0.9% | 0.9% |
| SpaceInvaders | 10.1% | 1.9% | 0.0% | 4.1% | 6.2% | 5.0% | 2.1% | 1.8% | 1.7% |
| Tutankham | 18.7% | 0.0% | 0.0% | 0.4% | 0.6% | 0.6% | 0.6% | 0.5% | 0.4% |
| WizardOfWor | 2.2% | 0.0% | 0.0% | 0.4% | 0.5% | 0.5% | 5.5% | 5.2% | 15.0% |
| Zaxxon | 2.3% | 0.8% | 53.9% | 7.9% | 9.3% | 8.4% | 3.7% | 3.5% | 4.3% |

Figure 7: Option usage (left) and specialization across Atari games (right) of a Hierarchical Option-Critic architecture with $N = 3$ and 3 options at each layer trained in the many task learning setting.

see that a hierarchy of options imposes structure in the space of options. For example, when $o^1 = 1$ or $o^1 = 2$ the low level options tend to focus on different situations within the same games.

## 6 Conclusion

In this work we propose the first policy gradient theorems to optimize an arbitrarily deep hierarchy of options to maximize the expected discounted return. Moreover, we have proposed a particular hierarchical option-critic architecture that is the first general purpose reinforcement learning architecture to successfully learn options from data with more than two abstraction levels. We have conducted extensive empirical evaluation in the tabular and deep non-linear function approximation settings. In all cases we found that, for significantly complex problems, reasoning with more than two levels of abstraction can be beneficial for learning. While the performance of the hierarchical option-critic architecture is impressive, we envision our proposed policy gradient theorems eventually transcending it in overall impact. Although the architectures we explore in this paper have a fixed structure and fixed depth of abstraction for simplicity, the underlying theorems can also guide learning for much more dynamic architectures that we hope to explore in future work.

## Acknowledgements

The authors thank Murray Campbell, Xiaoxiao Guo, Ignacio Cases, and Tim Klinger for fruitful discussions that helped shape this work.

## Footnotes

[1]Note that when no options terminate, as in the first term in equation (10), the lowest level option does not terminate and thus no higher level options have the opportunity to terminate.

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
