[Supplementary Material]

# Learning Abstract Options

**Matthew Riemer, Miao Liu, and Gerald Tesauro**
IBM Research
T.J. Watson Research Center, Yorktown Heights, NY
{mdriemer, miao.liu1, gtesauro}@us.ibm.com

# 1 Derivation of Generalized Policy Gradient and Termination Gradient Theorems

## 1.1 The Derivation of U

To help explain the meaning and derivation of equation (10), we separate the expression into four primary terms. The first term is applicable for $N \geq 1$ and represents the expected return from cases where no options terminate. The second term is applicable for $N \geq 2$ and represents the expected return from cases where every option terminates. The third and fourth terms are applicable for $N \geq 3$ and represent the expected return from cases where some options terminate.

We will first discuss how to estimate the return when there are no terminated options. In this case we simply use our estimate of the value of the current state following the current options if there are any. As we are computing the expectation, we also multiply this term by its likelihood of happening which is equal to the probability that the lowest level option policy does not terminate. When $N = 1$ we can consider the termination probability of the current policy as zero and the current option context to be empty. As such, we estimate the value function upon arrival as $V_\Omega(s)$ as we do for actor-critic policy gradients.

Next we turn our attention to estimating the return when all options are terminated. This can be approximated using our estimate of the return given the state $V_\Omega(s)$. The likelihood of this happening is equal to the conditional likelihood of options terminating at every level of abstraction we are modeling. When $N = 2$, equation (10) simplifies to equation (3). This expression is precisely the option value function upon arrival of the option-critic framework derived in [1].

The final quantity we will estimate bridges the gap to cases where only some options terminate. This situation has not been explored by other work on option learning as it only arises for situations with at least $N = 3$ hierarchical levels of planning. The case where some (but not all) options terminate arises when a series of low level options terminate while a high level option does not terminate. For a given level of abstraction, we can analyze the likelihood that at each level the lower level options terminate while the current does not. In such a case, we multiply this likelihood by the value one level more abstract than the current option hierarchy level. For convenience in our derivation, we split our notation for this quantity into two separate terms accounting explicitly for the case when only lower level options terminate.

## 1.2 Generalized Markov Chain and Augmented Process

We must establish the Markov chain along which we can measure performance for options with $N$ levels of abstraction. The natural approach is to consider the chain defined in the augmented state space because state and active option based tuples now play the role of regular states in a usual Markov chain. If options $o_t^{1:N-1}$ have been initiated or are executing at time $t$ in state $s_t$, then the probability of transitioning to $(s_{t+1}, o_{t+1}^{1:\ell-1})$ in one step is:

$$P(s_{t+1}, o_{t+1}^{1:\ell-1} | s_t, o_t^{1:N-1}) = \sum_{o_t^N} \pi_{\theta^N}^N(o_t^N | s_t, o_t^{1:N-1}) P(s_{t+1} | s_t, o_t^N)[$$

$$\underbrace{(1 - \beta_{\phi^{N-1}}^{N-1}(s_{t+1}, o_t^{1:N-1})) \mathbf{1}_{o_{t+1}^{1:\ell-1} = o_t^{1:\ell-1}}}_{\text{none terminate}} +$$

$$\underbrace{\sum_{q=N-1}^{\ell} (1 - \beta_{\phi^{q-1}}^{q-1}(s_{t+1}, o_t^{1:q-1})) \prod_{z=N-1}^{q} \beta_{\phi^z}^z(s_{t+1}, o_t^{1:z}) \mathbf{1}_{o_{t+1}^{1:\ell-1} = o_t^{1:\ell-1}}}_{\text{only lower level options terminate}} +$$

(1)

$$\underbrace{\prod_{j=N-1}^{1} \beta_{\phi^j}^j(s_{t+1}, o_t^{1:j}) \prod_{v=\ell-1}^{1} \pi_{\theta^v}^v(o_{t+1}^v | s_{t+1}, o_t^{1:v-1})}_{\text{all options terminate}} +$$

$$\underbrace{\sum_{i=1}^{\ell-2} (1 - \beta_{\phi^i}^i(s_{t+1}, o_t^{1:i})) \prod_{k=i+1}^{N-1} \beta_{\phi^k}^k(s_{t+1}, o_t^{1:k}) \prod_{p=i+1}^{\ell-1} \pi_{\theta^p}^p(o_{t+1}^p | s_{t+1}, o_t^{1:p-1})}_{\text{some relevant higher level options terminate}}.$$

where primitive actions are $o^N$. Like the Markov chain derived for the option critic architecture [1], the process given by equation (1) is homogeneous. Additionally, when options are available at every state, the process is ergodic with the existance of a unique stationary distribution over the augmented state space tuples.

We continue by presenting an extension of results about augmented processes used for derivation of learning algorithms in [1] to an option hierarchy with $N$ levels of abstraction. If options $o_t^{1:N-1}$ have been initiated or are executing at time $t$, then the discounted probability of transitioning to $(s_{t+1}, o_{t+1}^{1:\ell-1})$ where $\ell \leq N$ is:

$$P_\gamma^{(1)}(s_{t+1}, o_{t+1}^{1:\ell-1} | s_t, o_t^{1:N-1}) = \sum_{o_t^N} \pi_{\theta^N}^N(o_t^N | s_t, o_t^{1:N-1}) \gamma P(s_{t+1} | s_t, o_t^N)[$$

$$\underbrace{(1 - \beta_{\phi^{N-1}}^{N-1}(s_{t+1}, o_t^{1:N-1})) \mathbf{1}_{o_{t+1}^{1:\ell-1} = o_t^{1:\ell-1}}}_{\text{none terminate}} +$$

$$\underbrace{\sum_{q=N-1}^{\ell} (1 - \beta_{\phi^{q-1}}^{q-1}(s_{t+1}, o_t^{1:q-1})) \prod_{z=N-1}^{q} \beta_{\phi^z}^z(s_{t+1}, o_t^{1:z}) \mathbf{1}_{o_{t+1}^{1:\ell-1} = o_t^{1:\ell-1}}}_{\text{only lower level options terminate}} +$$

(2)

$$\underbrace{\prod_{j=N-1}^{1} \beta_{\phi^j}^j(s_{t+1}, o_t^{1:j}) \prod_{v=\ell-1}^{1} \pi_{\theta^v}^v(o_{t+1}^v | s_{t+1}, o_t^{1:v-1})}_{\text{all options terminate}} +$$

$$\underbrace{\sum_{i=1}^{\ell-2} (1 - \beta_{\phi^i}^i(s_{t+1}, o_t^{1:i})) \prod_{k=i+1}^{N-1} \beta_{\phi^k}^k(s_{t+1}, o_t^{1:k}) \prod_{p=i+1}^{\ell-1} \pi_{\theta^p}^p(o_{t+1}^p | s_{t+1}, o_t^{1:p-1})}_{\text{some relevant higher level options terminate}}.$$

As such, when we condition the process from $(s_t, o_{t-1}^{1:N-1})$, the discounted probability of transitioning to $(s_{t+1}, o_t^{1:\ell-1})$ is:

$$P_\gamma^{(1)}(s_{t+1}, o_t^{1:\ell-1}|s_t, o_{t-1}^{1:N-1}) = \sum_{o_t^N} \pi_{\theta^N}^N(o_t^N|s_t, o_t^{1:N-1})\gamma P(s_{t+1}|s_t, o_t^N)[$$

$$\underbrace{(1-\beta_{\phi^{N-1}}^{N-1}(s_{t+1}, o_{t-1}^{1:N-1}))\mathbf{1}_{o_t^{1:\ell-1}=o_{t-1}^{1:\ell-1}}}_{\text{none terminate}} +$$

$$\underbrace{\sum_{q=N-1}^{\ell}(1-\beta_{\phi^{q-1}}^{q-1}(s_{t+1}, o_{t-1}^{1:q-1}))\prod_{z=N-1}^{q}\beta_{\phi^z}^z(s_{t+1}, o_{1:t-1}^z)\mathbf{1}_{o_t^{1:\ell-1}=o_{t-1}^{1:\ell-1}}}_{\text{only lower level options terminate}} +$$

$$\underbrace{\prod_{j=N-1}^{1}\beta_{\phi^j}^j(s_{t+1}, o_{t-1}^{1:j})\prod_{v=\ell-1}^{1}\pi_{\theta^v}^v(o_t^v|s_{t+1}, o_{t-1}^{1:v-1})}_{\text{all options terminate}} +$$

$$\underbrace{\sum_{i=1}^{\ell-2}(1-\beta_{\phi^i}^i(s_{t+1}, o_{t-1}^{1:i}))\prod_{k=i+1}^{N-1}\beta_{\phi^k}^k(s_{t+1}, o_{t-1}^{1:k})\prod_{p=i+1}^{\ell-1}\pi_{\theta^p}^p(o_t^p|s_{t+1}, o_{t-1}^{1:p-1})}_{\text{some relevant higher level options terminate}}].$$

(3)

This definition will be very useful later for our derivation of the hierarchical intra-option policy gradient. However, for the derivation of the hierarchical termination gradient theorem we should reformulate the discounted probability of transitioning to $(s_{t+1}, o_t^{1:\ell})$ from the view of the termination policy at abstraction level $\ell$ explicitly separating out terms that depend on $\phi^\ell$:

$$P_\gamma^{(1)}(s_{t+1}, o_t^{1:\ell}|s_t, o_{t-1}^{1:N-1}) = \sum_{o_t^N}\pi_{\theta^N}^N(o_t^N|s_t, o_t^{1:N-1})\gamma P(s_{t+1}|s_t, o_t^N)[$$

$$\underbrace{(1-\beta_{\phi^{N-1}}^{N-1}(s_{t+1}, o_{t-1}^{1:N-1}))\mathbf{1}_{o_t^{1:\ell}=o_{t-1}^{1:\ell}}}_{\text{none terminate}} + \underbrace{\sum_{q=N-1}^{\ell+2}(1-\beta_{\phi^{q-1}}^{q-1}(s_{t+1}, o_{t-1}^{1:q-1}))\prod_{z=N-1}^{q}\beta_{\phi^z}^z(s_{t+1}, o_{t-1}^{1:z})\mathbf{1}_{o_t^{1:\ell}=o_{t-1}^{1:\ell}}}_{\text{only lower level options terminate}} +$$

$$\underbrace{(1-\beta_{\phi^\ell}^\ell(s_{t+1}, o_{t-1}^{1:\ell}))\prod_{z=N-1}^{\ell+1}\beta_{\phi^z}^z(s_{t+1}, o_{t-1}^{1:z})\mathbf{1}_{o_t^{1:\ell}=o_{t-1}^{1:\ell}}}_{\ell+1 \text{ terminates and } \ell \text{ does not}} + \underbrace{\prod_{j=N-1}^{1}\beta_{\phi^j}^j(s_{t+1}, o_{t-1}^{1:j})\prod_{v=\ell}^{1}\pi_{\theta^v}^v(o_t^v|s_{t+1}, o_{t-1}^{1:v-1})}_{\text{all options terminate}} +$$

$$\underbrace{\sum_{i=1}^{\ell-1}(1-\beta_{\phi^i}^i(s_{t+1}, o_{t-1}^{1:i}))\prod_{k=i+1}^{N-1}\beta_{\phi^k}^k(s_{t+1}, o_{t-1}^{1:k})\prod_{p=i+1}^{\ell}\pi_{\theta^p}^p(o_t^p|s_{t+1}, o_{t-1}^{1:p-1})}_{\text{some relevant higher level options terminate}}].$$

(4)

The $k$-step discounted probabilities can more generally be expressed recursively:

$$P_\gamma^{(k)}(s_{t+k}, o_{t+k}^{1:\ell-1}|s_t, o_t^{1:N-1}) =$$

$$\sum_{s_{t+1}}\sum_{o_{t+1}^1}\cdots\sum_{o_{t+1}^{N-1}}[P_\gamma^{(1)}(s_{t+1}, o_{t+1}^{1:N-1}|s_t, o_t^{1:N-1})P_\gamma^{(k-1)}(s_{t+k-1}, o_{t+k}^{1:\ell-1}|s_{t+1}, o_{t+1}^{1:N-1})].$$

(5)

Or rather conditioning on $t-1$ as in equation (3):

$$P_\gamma^{(k)}(s_{t+k}, o_{t+k-1}^{1:\ell-1}|s_t, o_{t-1}^{1:N-1}) =$$

$$\sum_{s_{t+1}}\sum_{o_t^1}\cdots\sum_{o_t^{1:N-1}}[P_\gamma^{(1)}(s_{t+1}, o_t^{1:N-1}|s_t, o_{t-1}^{1:N-1})P_\gamma^{(k-1)}(s_{t+k-1}, o_{t+k-1}^{1:\ell-1}|s_{t+1}, o_t^{1:N-1})].$$

(6)

## 1.3 Proof of the Hierarchical Intra-Option Policy Gradient Theorem

Taking the gradient of the value function with an augmented state space:

$$\frac{\partial Q_\Omega(s, o^{1:\ell-1})}{\partial \theta^\ell} = \frac{\partial}{\partial \theta^\ell} \sum_{o^\ell} \pi^\ell_{\theta^\ell}(o^\ell | s, o^{1:\ell-1}) Q_U(s, o^{1:\ell})$$

$$= \sum_{o^\ell} \left( \frac{\partial \pi^\ell_{\theta^\ell}(o^\ell | s, o^{1:\ell-1})}{\partial \theta^\ell} Q_U(s, o^{1:\ell}) + \pi^\ell_{\theta^\ell}(o^\ell | s, o^{1:\ell-1}) \frac{\partial Q_U(s, o^{1:\ell})}{\partial \theta^\ell} \right) \quad (7)$$

Then substituting in equation 9 with the assumption that $\theta^\ell$ only appears in the intra-option policy at level $\ell$ and not in any policy at another level or in the termination function:

$$\frac{\partial Q_\Omega(s, o^{1:\ell-1})}{\partial \theta^\ell} = \sum_{o^\ell} \left( \frac{\partial \pi^\ell_{\theta^\ell}(o^\ell | s, o^{1:\ell-1})}{\partial \theta^\ell} Q_U(s, o^{1:\ell}) + \pi^\ell_{\theta^\ell}(o^\ell | s, o^{1:\ell-1}) \gamma \sum_{s'} P(s' | s, o^{1:\ell}) \frac{\partial U(s', o^{1:\ell-1})}{\partial \theta^\ell} \right) \quad (8)$$

where $P(s' | s, o^{1:\ell})$ is the probability of transitioning to a state based on the augmented state space $(s, o^{1:\ell})$ considering primitive actions $o^N$:

$$P(s' | s, o^{1:\ell}) = \sum_{o^N} \cdots \sum_{o^{\ell+1}} P(s' | s, o^N) \prod_{j=\ell+1}^{N} \pi^j(o^j | s, o^{1:j-1}). \quad (9)$$

We continue by computing the gradient with respect to $U$ again assuming that $\theta^\ell$ only appears in the intra-option policy at level $\ell$ and not in any policy at another level or in the termination function:

$$\frac{\partial U(s', o^{1:\ell-1})}{\partial \theta^\ell} = \underbrace{(1 - \beta^{N-1}_{\phi^{N-1}}(s', o^{1:N-1})) \frac{\partial Q_\Omega(s', o^{1:\ell-1})}{\partial \theta^\ell}}_{\text{none terminate}} + \underbrace{\frac{\partial V_\Omega(s')}{\partial \theta^\ell} \prod_{j=N-1}^{1} \beta^j_{\phi^j}(s', o^{1:j})}_{\text{all options terminate}} +$$

$$\underbrace{\frac{\partial Q_\Omega(s', o^{1:\ell-1})}{\partial \theta^\ell} \sum_{q=N-1}^{\ell} (1 - \beta^{q-1}_{\phi^{q-1}}(s', o^{1:q-1})) \prod_{z=N}^{q} \beta^z_{\phi^z}(s', o^{1:z})}_{\text{only lower level options terminate}} +$$

$$\underbrace{\sum_{i=1}^{\ell-2} (1 - \beta^i_{\phi^i}(s', o^{1:i})) \frac{\partial Q_\Omega(s', o^{1:i})}{\partial \theta^\ell} \prod_{k=i+1}^{N-1} \beta^k_{\phi^k}(s', o^{1:k})}_{\text{some relevant higher level options terminate}} \quad (10)$$

Next we integrate out the lower level options so that each term is operating in the same augmented state space:

$$\frac{\partial U(s', o^{1:\ell-1})}{\partial \theta^\ell} = \underbrace{(1 - \beta^{N-1}_{\phi^{N-1}}(s', o^{1:N-1})) \frac{\partial Q_\Omega(s', o^{1:\ell-1})}{\partial \theta^\ell}}_{\text{none terminate}} +$$

$$\underbrace{\sum_{q=N-1}^{\ell} (1 - \beta^{q-1}_{\phi^{q-1}}(s', o^{1:q-1})) \prod_{z=N-1}^{q} \beta^z_{\phi^z}(s', o^{1:z}) \frac{\partial Q_\Omega(s', o^{1:\ell-1})}{\partial \theta^\ell}}_{\text{only lower level options terminate}} +$$

$$\underbrace{\prod_{j=N-1}^{1} \beta^j_{\phi^j}(s', o^{1:j}) \sum_{o'^1} \cdots \sum_{o'^{\ell-1}} \prod_{v=\ell-1}^{1} \pi^v_{\theta^v}(o'^v | s', o'^{1:v-1}) \frac{\partial Q_\Omega(s', o'^{1:\ell-1})}{\partial \theta^\ell}}_{\text{all options terminate}} + \quad (11)$$

$$\underbrace{\sum_{i=1}^{\ell-2} (1 - \beta^i_{\phi^i}(s', o^{1:i})) \prod_{k=i+1}^{N-1} \beta^k_{\phi^k}(s', o^{1:k}) \sum_{o'^{i+1}} \cdots \sum_{o'^{\ell-1}} \prod_{p=i+1}^{\ell-1} \pi^p_{\theta^p}(o'^p | s', o'^{1:p-1}) \frac{\partial Q_\Omega(s', o'^{1:\ell-1})}{\partial \theta^\ell}}_{\text{some relevant higher level options terminate}}$$

We can then simplify our expression:

$$\frac{\partial U(s', o^{1:\ell-1})}{\partial \theta^\ell} = \sum_{o'^1} \cdots \sum_{o'^{\ell-1}} [\underbrace{(1 - \beta^{N-1}_{\phi^{N-1}}(s', o^{1:N-1}))\mathbf{1}_{o'^{1:\ell-1}=o^{1:\ell-1}}}_{\text{none terminate}} +$$

$$\underbrace{\sum_{q=N-1}^{\ell}(1 - \beta^{q-1}_{\phi^{q-1}}(s', o^{1:q-1}))\prod_{z=N}^{q}\beta^{z}_{\phi^z}(s', o^{1:z})\mathbf{1}_{o'^{1:\ell-1}=o^{1:\ell-1}}}_{\text{only lower level options terminate}} +$$

$$\underbrace{\prod_{j=N-1}^{1}\beta^{j}_{\phi^j}(s', o^{1:j})\prod_{v=\ell-1}^{1}\pi^{v}_{\theta^{v}}(o'^{v}|s', o'^{1:v-1})}_{\text{all options terminate}} + \tag{12}$$

$$\underbrace{\sum_{i=1}^{\ell-2}(1 - \beta^{i}_{\phi^i}(s', o^{1:i}))\prod_{k=i+1}^{N-1}\beta^{k}_{\phi^k}(s', o^{1:k})\prod_{p=i+1}^{\ell-1}\pi^{p}_{\theta^{p}}(o'^{p}|s', o'^{1:p-1})}_{\text{some relevant higher level options terminate}}]\frac{\partial Q_\Omega(s', o'^{1:\ell-1})}{\partial \theta^\ell},$$

We proceed by substituting (12) into (8):

$$\frac{\partial Q_\Omega(s, o^{1:\ell-1})}{\partial \theta^\ell} = \sum_{o^\ell}\left(\frac{\partial \pi^{\ell}_{\theta^\ell}(o^\ell|s, o^{1:\ell-1})}{\partial \theta^\ell}Q_U(s, o^{1:\ell})+\right.$$

$$\pi^{\ell}_{\theta^\ell}(o^\ell|s, o^{1:\ell-1})\gamma\sum_{s'}P(s'|s, o^{1:\ell})\sum_{o'^1}\cdots\sum_{o'^{\ell-1}}[\underbrace{(1 - \beta^{N-1}_{\phi^{N-1}}(s', o^{1:N-1}))\mathbf{1}_{o'^{1:\ell-1}=o^{1:\ell-1}}}_{\text{none terminate}} +$$

$$\underbrace{\sum_{q=N-1}^{\ell}(1 - \beta^{q-1}_{\phi^{q-1}}(s', o^{1:q-1}))\prod_{z=N}^{q}\beta^{z}_{\phi^z}(s', o^{1:z})\mathbf{1}_{o'^{1:\ell-1}=o^{1:\ell-1}}}_{\text{only lower level options terminate}} +$$

$$\underbrace{\prod_{j=N-1}^{1}\beta^{j}_{\phi^j}(s', o^{1:j})\prod_{v=\ell-1}^{1}\pi^{v}_{\theta^{v}}(o'^{v}|s', o'^{1:v-1})}_{\text{all options terminate}} + \tag{13}$$

$$\underbrace{\sum_{i=1}^{\ell-2}(1 - \beta^{i}_{\phi^i}(s', o^{1:i}))\prod_{k=i+1}^{N-1}\beta^{k}_{\phi^k}(s', o^{1:k})\prod_{p=i+1}^{\ell-1}\pi^{p}_{\theta^{p}}(o'^{p}|s', o'^{1:p-1})}_{\text{some relevant higher level options terminate}}]\frac{\partial Q_\Omega(s', o'^{1:\ell-1})}{\partial \theta^\ell}$$

This yields a recursion, which can be further simplified to:

$$\frac{\partial Q_\Omega(s, o^{1:\ell-1})}{\partial \theta^\ell} =$$

$$\sum_{o^\ell}\frac{\partial \pi^{\ell}_{\theta^\ell}(o^\ell|s, o^{1:\ell-1})}{\partial \theta^\ell}Q_U(s, o^{1:\ell}) + \sum_{s'}\sum_{o'^1}\cdots\sum_{o'^{\ell-1}}P^{(1)}_{\gamma}(s', o'^{1:\ell-1}|s, o^{1:N-1})\frac{\partial Q_\Omega(s', o'^{1:\ell-1})}{\partial \theta^\ell} \tag{14}$$

Considering the previous remarks about augmented processes and substituting in equation (3), this expression becomes:

$$\frac{\partial Q_\Omega(s, o^{1:\ell-1})}{\partial \theta^\ell} = \sum_{k=0}^{\infty}\sum_{s', o'^{1:\ell-1}}P^{(k)}_{\gamma}(s', o'^{1:\ell-1}|s, o^{1:N-1})\sum_{o^\ell}\frac{\partial \pi^{\ell}_{\theta^\ell}(o'^\ell|s', o'^{1:\ell-1})}{\partial \theta^\ell}Q_U(s', o'^{1:\ell}) \tag{15}$$

The gradient of the expected discounted return with respect to $\theta^\ell$ is then:

$$
\begin{aligned}
\frac{\partial Q_\Omega(s_0, o_0^{1:\ell-1})}{\partial \theta^\ell} &= \sum_{s, o^{1:\ell-1}} \sum_{k=0}^{\infty} P_\gamma^{(k)}(s, o^{1:\ell-1} | s_0, o_0^{1:N-1}) \sum_{o^\ell} \frac{\partial \pi_{\theta^\ell}^\ell(o^\ell | s, o^{1:\ell-1})}{\partial \theta^\ell} Q_U(s, o^{1:\ell}) \\
&= \sum_{s, o^{1:\ell-1}} \mu_\Omega(s, o^{1:\ell-1} | s_0, o_0^{1:N-1}) \sum_{o^\ell} \frac{\partial \pi_{\theta^\ell}^\ell(o^\ell | s, o^{1:\ell-1})}{\partial \theta^\ell} Q_U(s, o^{1:\ell}).
\end{aligned}
\tag{16}
$$

## 1.4 Proof of the Hierarchical Termination Gradient Theorem

The expected sum of discounted rewards originating from augmented state $(s_1, o_0^{1:N-1})$ is defined as:

$$
U(s_1, o_0^{1:N-1}) = \mathbb{E}\left[\sum_{t=1}^{\infty} \gamma^{t-1} r_t | s_1, o_0^{1:N-1}\right]
\tag{17}
$$

We start by reformulating $U$ from equation (10) at level of abstraction $\ell$ rather than $\ell - 1$ as follows:

$$
\begin{aligned}
U(s', o^{1:\ell}) = \underbrace{(1 - \beta_{\phi^{N-1}}^{N-1}(s', o^{1:N-1})) Q_\Omega(s', o^{1:\ell})}_{\text{none terminate}} + \underbrace{V_\Omega(s) \prod_{j=N-1}^{1} \beta_{\phi^j}^j(s', o^{1:j})}_{\text{all options terminate}} + \\
\underbrace{Q_\Omega(s', o^{1:\ell}) \sum_{q=N-1}^{\ell+1} (1 - \beta_{\phi^{q-1}}^{q-1}(s', o^{1:q-1})) \prod_{z=N-1}^{q} \beta_{\phi^z}^z(s', o^{1:z})}_{\text{only lower level options terminate}} + \\
\underbrace{\sum_{i=1}^{\ell-1} (1 - \beta_{\phi^i}^i(s', o^{1:i})) Q_\Omega(s', o^{1:i}) \prod_{k=i+1}^{N-1} \beta_{\phi^k}^k(s', o^{1:k})}_{\text{some relevant higher level options terminate}}
\end{aligned}
\tag{18}
$$

As we will be interested in analyzing this expression with respect to $\phi^\ell$, we separate the term where only lower level options terminate into two separate terms. In the special case where $\ell + 1$ terminates and $\ell$ does not, we still utilize $\phi^\ell$ even though it did not terminate:

$$
\begin{aligned}
U(s', o^{1:\ell}) = \underbrace{(1 - \beta_{\phi^{N-1}}^{N-1}(s', o^{1:N-1})) Q_\Omega(s', o^{1:\ell})}_{\text{none terminate}} + \underbrace{V_\Omega(s) \prod_{j=N-1}^{1} \beta_{\phi^j}^j(s', o^{1:j})}_{\text{all options terminate}} + \\
\underbrace{Q_\Omega(s', o^{1:\ell}) \sum_{q=N-1}^{\ell+2} (1 - \beta_{\phi^{q-1}}^{q-1}(s', o^{1:q-1})) \prod_{z=N-1}^{q} \beta_{\phi^z}^z(s', o^{1:z})}_{\text{only lower level options than } \ell+1 \text{ terminate}} + \\
\underbrace{Q_\Omega(s', o^{1:\ell})(1 - \beta_{\phi^\ell}^\ell(s', o^{1:\ell})) \prod_{z=N-1}^{\ell+1} \beta_{\phi^z}^z(s', o^{1:z})}_{\ell+1 \text{ terminates and } \ell \text{ does not}} + \underbrace{\sum_{i=1}^{\ell-1} (1 - \beta_{\phi^i}^i(s', o^{1:i})) Q_\Omega(s', o^{1:i}) \prod_{k=i+1}^{N-1} \beta_{\phi^k}^k(s', o^{1:k})}_{\text{some relevant higher level options terminate}}
\end{aligned}
\tag{19}
$$

The original expression of $U$ was more useful for the gradient with respect to $\theta^\ell$, which does not depend on this case. The gradient of $U$ with respect to $\phi^\ell$ is then:

$$
\frac{\partial U(s', o^{1:\ell})}{\partial \phi^\ell} = \underbrace{V_\Omega(s) \frac{\partial \beta_{\phi^\ell}^\ell(s', o^{1:\ell})}{\partial \phi^\ell}[\prod_{j=N-1}^{\ell+1} \beta_{\phi^j}^j(s', o^{1:j})][\prod_{j=\ell-1}^{1} \beta_{\phi^j}^j(s', o^{1:j})]}_{(1)\text{ all options terminate}} +
$$

$$
\underbrace{Q_\Omega(s', o^{1:\ell})(-\frac{\partial \beta_{\phi^\ell}^\ell(s', o^{1:\ell})}{\partial \phi^\ell}) \prod_{z=N-1}^{\ell+1} \beta_{\phi^z}^z(s', o^{1:z})}_{(2)\ \ell+1\text{ terminates and }\ell\text{ does not}} +
$$

$$
\underbrace{\sum_{i=1}^{\ell-1}(1 - \beta_{\phi^i}^i(s', o^{1:i}))Q_\Omega(s', o^{1:i}) \frac{\partial \beta_{\phi^\ell}^\ell(s', o^{1:\ell})}{\partial \phi^\ell}[\prod_{k=i+1}^{\ell-1} \beta_{\phi^k}^k(s', o^{1:k})][\prod_{k=\ell+1}^{N-1} \beta_{\phi^k}^k(s', o^{1:k})]}_{(3)\text{ some relevant higher level options terminate}} +
$$

$$
\underbrace{(1 - \beta_{\phi^{N-1}}^{N-1}(s', o^{1:N-1})) \frac{\partial Q_\Omega(s', o^{1:\ell})}{\partial \phi^\ell}}_{(4)\text{ none terminate}} + \quad (20)
$$

$$
+ \underbrace{\frac{\partial V_\Omega(s)}{\partial \phi^\ell} \prod_{j=N-1}^{1} \beta_{\phi^j}^j(s', o^{1:j})}_{(5)\text{ all options terminate}} + \underbrace{\frac{\partial Q_\Omega(s', o^{1:\ell})}{\partial \phi^\ell} \sum_{q=N-1}^{\ell+2}(1 - \beta_{\phi^{q-1}}^{q-1}(s', o^{1:q-1})) \prod_{z=N-1}^{q} \beta_{\phi^z}^z(s', o^{1:z})}_{(6)\text{ only lower level options than }\ell+1\text{ terminate}} +
$$

$$
\underbrace{\frac{\partial Q_\Omega(s', o^{1:\ell})}{\partial \phi^\ell}(1 - \beta_{\phi^\ell}^\ell(s', o^{1:\ell})) \prod_{z=N-1}^{\ell+2} \beta_{\phi^z}^z(s', o^{1:z})}_{(7)\ \ell+1\text{ terminates and }\ell\text{ does not}} +
$$

$$
\underbrace{\sum_{i=1}^{\ell-1}(1 - \beta_{\phi^i}^i(s', o^{1:i})) \frac{\partial Q_\Omega(s', o^{1:i})}{\partial \phi^\ell} \prod_{k=i+1}^{N-1} \beta_{\phi^k}^k(s', o^{1:k})}_{(8)\text{ some relevant higher level options terminate}}
$$

Merging the first three terms as well as the 6th and 7th terms:

$$
\frac{\partial U(s', o^{1:\ell})}{\partial \phi^\ell} = \prod_{j=N-1}^{\ell+1} \beta_{\phi^j}^j(s', o^{1:j}) \frac{\partial \beta_{\phi^\ell}^\ell(s', o^{1:\ell})}{\partial \phi^\ell}[\underbrace{-Q_\Omega(s', o^{1:\ell})}_{\ell+1\text{ terminates and }\ell\text{ does not}} +
$$

$$
\underbrace{V_\Omega(s)[\prod_{j=\ell-1}^{1} \beta_{\phi^j}^j(s', o^{1:j})]}_{\text{all options terminate}} + \underbrace{\sum_{i=1}^{\ell-1}(1 - \beta_{\phi^i}^i(s', o^{1:i}))Q_\Omega(s', o^{1:i})[\prod_{k=i+1}^{\ell-1} \beta_{\phi^k}^k(s', o^{1:k})]]}_{\text{some relevant higher level options terminate}}
$$

$$
+ \underbrace{(1 - \beta_{\phi^{N-1}}^{N-1}(s', o^{1:N-1})) \frac{\partial Q_\Omega(s', o^{1:\ell})}{\partial \phi^\ell}}_{\text{none terminate}} + \underbrace{\frac{\partial V_\Omega(s)}{\partial \phi^\ell} \prod_{j=N-1}^{1} \beta_{\phi^j}^j(s', o^{1:j})}_{\text{all options terminate}} + \quad (21)
$$

$$
\underbrace{\frac{\partial Q_\Omega(s', o^{1:\ell})}{\partial \phi^\ell} \sum_{q=N-1}^{\ell+1}(1 - \beta_{\phi^{q-1}}^{q-1}(s', o^{1:q-1})) \prod_{z=N-1}^{q} \beta_{\phi^z}^z(s', o^{1:z})}_{\text{only lower level options terminate}} +
$$

$$
\underbrace{\sum_{i=1}^{\ell-1}(1 - \beta_{\phi^i}^i(s', o^{1:i})) \frac{\partial Q_\Omega(s', o^{1:i})}{\partial \phi^\ell} \prod_{k=i+1}^{N-1} \beta_{\phi^k}^k(s', o^{1:k})}_{\text{some relevant higher level options terminate}}
$$

We define the probability weighted advantage of not terminating $A_\Omega$ as:

$$A_\Omega(s', o^{1:\ell}) = Q_\Omega(s', o^{1:\ell}) - V_\Omega(s)[\prod_{j=\ell-1}^{1} \beta_{\phi^j}^j(s', o^{1:j})] - \sum_{i=1}^{\ell-1}(1 - \beta_{\phi^i}^i(s', o^{1:i}))Q_\Omega(s', o^{1:i})[\prod_{k=i+1}^{\ell-1} \beta_{\phi^k}^k(s', o^{1:k})]$$

(22)

We proceed to substitute equation (22) into equation (21):

$$\frac{\partial U(s', o^{1:\ell})}{\partial \phi^\ell} = -\prod_{j=N-1}^{\ell+1} \beta_{\phi^j}^j(s', o^{1:j})\frac{\partial \beta_{\phi^\ell}^\ell(s', o^{1:\ell})}{\partial \phi^\ell}A_\Omega(s', o^{1:\ell})$$

$$+\underbrace{(1 - \beta_{\phi^{N-1}}^{N-1}(s', o^{1:N-1}))\frac{\partial Q_\Omega(s', o^{1:\ell})}{\partial \phi^\ell}}_{\text{(1) none terminate}} + \underbrace{\frac{\partial V_\Omega(s)}{\partial \phi^\ell}\prod_{j=N-1}^{1} \beta_{\phi^j}^j(s', o^{1:j})}_{\text{(2) all options terminate}} +$$

$$\underbrace{\frac{\partial Q_\Omega(s', o^{1:\ell})}{\partial \phi^\ell}\sum_{q=N-1}^{\ell+1}(1 - \beta_{\phi^{q-1}}^{q-1}(s', o^{1:q-1}))\prod_{z=N-1}^{q} \beta_{\phi^z}^z(s', o^{1:z})}_{\text{(3) only lower level options terminate}} +$$

(23)

$$\underbrace{\sum_{i=1}^{\ell-1}(1 - \beta_{\phi^i}^i(s', o^{1:i}))\frac{\partial Q_\Omega(s', o^{1:i})}{\partial \phi^\ell}\prod_{k=i+1}^{N-1} \beta_{\phi^k}^k(s', o^{1:k})]}_{\text{(4) some relevant higher level options terminate}}$$

Next we integrate out our last three terms so that they are in terms of a common derivative:

$$\frac{\partial U(s', o^{1:\ell})}{\partial \phi^\ell} = -\prod_{j=N-1}^{\ell+1} \beta_{\phi^j}^j(s', o^{1:j})\frac{\partial \beta_{\phi^\ell}^\ell(s', o^{1:\ell})}{\partial \phi^\ell}A_\Omega(s', o^{1:\ell})$$

$$+\underbrace{(1 - \beta_{\phi^{N-1}}^{N-1}(s', o^{1:N-1}))\frac{\partial Q_\Omega(s', o^{1:\ell})}{\partial \phi^\ell}}_{\text{none terminate}}$$

$$+\underbrace{\prod_{j=N-1}^{1} \beta_{\phi^j}^j(s', o^{1:j})\sum_{o'^1}...\sum_{o'^\ell}\prod_{v=\ell}^{1} \pi_{\theta^v}^v(o'^v|s', o'^{1:v-1})\frac{\partial Q_\Omega(s', o'^{1:\ell})}{\partial \phi^\ell}}_{\text{all options terminate}} +$$

(24)

$$\underbrace{\frac{\partial Q_\Omega(s', o^{1:\ell})}{\partial \phi^\ell}\sum_{q=N-1}^{\ell+1}(1 - \beta_{\phi^{q-1}}^{q-1}(s', o^{1:q-1}))\prod_{z=N-1}^{q} \beta_{\phi^z}^z(s', o^{1:z})}_{\text{only lower level options terminate}} +$$

$$\underbrace{\sum_{i=1}^{\ell-1}(1 - \beta_{\phi^i}^i(s', o^{1:i}))\frac{\partial Q_\Omega(s', o^{1:\ell})}{\partial \phi^\ell}\prod_{k=i+1}^{N-1} \beta_{\phi^k}^k(s', o^{1:k})\prod_{p=i+1}^{\ell} \pi_{\theta^p}^p(o'^p|s', o'^{1:p-1})]}_{\text{some relevant higher level options terminate}}$$

We can then simplify the expression:

$$
\frac{\partial U(s',o^{1:\ell})}{\partial \phi^\ell} = - \prod_{j=N-1}^{\ell+1} \beta_{\phi^j}^j(s',o^{1:j}) \frac{\partial \beta_{\phi^\ell}^\ell(s',o^{1:\ell})}{\partial \phi^\ell} A_\Omega(s',o^{1:\ell}) +
$$

$$
[\underbrace{(1-\beta_{\phi^{N-1}}^{N-1}(s',o^{1:N-1}))\mathbf{1}_{o'^{1:\ell}=o^{1:\ell}}}_{\text{none terminate}} + \underbrace{\prod_{j=N-1}^{1} \beta_{\phi^j}^j(s',o^{1:j}) \sum_{o'^1}...\sum_{o'^\ell} \prod_{v=\ell}^{1} \pi_{\theta^v}^v(o'^v|s',o'^{1:v-1})}_{\text{all options terminate}} +
$$

$$
\underbrace{\sum_{q=N-1}^{\ell+1} (1-\beta_{\phi^{q-1}}^{q-1}(s',o^{1:q-1})) \prod_{z=N-1}^{q} \beta_{\phi^z}^z(s',o^{1:z})\mathbf{1}_{o'^{1:\ell}=o^{1:\ell}}}_{\text{only lower level options terminate}} + \qquad (25)
$$

$$
\underbrace{\sum_{i=1}^{\ell-1}(1-\beta_{\phi^i}^i(s',o^{1:i})) \prod_{k=i+1}^{N-1} \beta_{\phi^k}^k(s',o^{1:k}) \prod_{p=i+1}^{\ell} \pi_{\theta^p}^p(o'^p|s',o'^{1:p-1})}_{\text{some relevant higher level options terminate}}] \frac{\partial Q_\Omega(s',o'^{1:\ell})}{\partial \phi^\ell}
$$

We now note that substituting equation (9) into equation (12) yields:

$$
\frac{\partial Q_\Omega(s,o^{1:\ell})}{\partial \phi^\ell} = \gamma P(s'|s,o^{1:\ell}) \frac{\partial U(s',o^{1:\ell})}{\partial \phi^\ell} \qquad (26)
$$

Substituting this expression into equation (25) we find that:

$$
\frac{\partial U(s',o^{1:\ell})}{\partial \phi^\ell} = - \prod_{j=N-1}^{\ell+1} \beta_{\phi^j}^j(s',o^{1:j}) \frac{\partial \beta_{\phi^\ell}^\ell(s',o^{1:\ell})}{\partial \phi^\ell} A_\Omega(s',o^{1:\ell}) +
$$

$$
[\underbrace{(1-\beta_{\phi^{N-1}}^{N-1}(s',o^{1:N-1}))\mathbf{1}_{o'^{1:\ell}=o^{1:\ell}}}_{\text{none terminate}} + \underbrace{\prod_{j=N-1}^{1} \beta_{\phi^j}^j(s',o^{1:j}) \sum_{o'^1}...\sum_{o'^\ell} \prod_{v=\ell}^{1} \pi_{\theta^v}^v(o'^v|s',o'^{1:v-1})}_{\text{all options terminate}} +
$$

$$
\underbrace{\sum_{q=N-1}^{\ell+1} (1-\beta_{\phi^{q-1}}^{q-1}(s',o^{1:q-1})) \prod_{z=N-1}^{q} \beta_{\phi^z}^z(s',o^{1:z})\mathbf{1}_{o'^{1:\ell}=o^{1:\ell}}}_{\text{only lower level options terminate}} + \qquad (27)
$$

$$
\underbrace{\sum_{i=1}^{\ell-1}(1-\beta_{\phi^i}^i(s',o^{1:i})) \prod_{k=i+1}^{N-1} \beta_{\phi^k}^k(s',o^{1:k}) \prod_{p=i+1}^{\ell} \pi_{\theta^p}^p(o'^p|s',o'^{1:p-1})}_{\text{some relevant higher level options terminate}}] \gamma P(s'|s,o^{1:\ell}) \frac{\partial U(s',o^{1:\ell})}{\partial \phi^\ell}
$$

Leveraging the augmented process structure and substituting in equation (4):

$$
\frac{\partial U(s',o^{1:\ell})}{\partial \phi^\ell} = - \prod_{i=\ell+1}^{N-1} \beta_{\phi^i}^i(s',o^{1:i}) \frac{\partial \beta_{\phi^\ell}^\ell(s',o^{1:\ell})}{\partial \phi^\ell} A_\Omega(s',o^{1:\ell}) + \sum_{s''}\sum_{o'^1}...\sum_{o'^\ell} P_\gamma^{(1)}(s'',o'^{1:\ell}|s,o^{1:N-1}) \frac{\partial U_\Omega(s'',o'^{1:\ell})}{\partial \phi^\ell}
$$

$$
= - \sum_{s'',o'^{1:\ell}} \sum_{k=0}^{\infty} P_\gamma^{(k)}(s'',o'^{1:\ell}|s,o^{1:N-1}) \prod_{i=\ell+1}^{N-1} \beta_{\phi^i}^i(s',o^{1:i}) \frac{\partial \beta_{\phi^\ell}^\ell(s',o^{1:\ell})}{\partial \phi^\ell} A_\Omega(s',o^{1:\ell}),
$$

$$
(28)
$$

We can then finally obtain that:

$$
\begin{aligned}
\frac{\partial U(s_1, o_0^{1:\ell})}{\partial \phi^\ell} &= - \sum_{s, o^{1:\ell}} \sum_{k=0}^{\infty} P_\gamma^{(k)}(s, o^{1:\ell}|s_1, o_0^{1:N-1}) \prod_{i=\ell+1}^{N-1} \beta_{\phi^i}^i(s, o^{1:i}) \frac{\partial \beta_{\phi^\ell}^\ell(s, o^{1:\ell})}{\partial \phi^\ell} A_\Omega(s, o^{1:\ell}) \\
&= - \sum_{s, o^{1:\ell}} \mu_\Omega(s, o^{1:\ell}|s_1, o_0^{1:N-1}) \prod_{i=\ell+1}^{N-1} \beta_{\phi^i}^i(s, o^{1:i}) \frac{\partial \beta_{\phi^\ell}^\ell(s, o^{1:\ell})}{\partial \phi^\ell} A_\Omega(s, o^{1:\ell}).
\end{aligned}
\tag{29}
$$

## 2 Additional Details for Experiments

In Algorithm 1 we provide a detailed algorithm for our learning policy in the tabular setting. This algorithm generalizes the one presented in [1] for option-critic learning to hierarchical option-critic learning with $N$ levels of abstraction. In Algorithm 2 we provide the same generalization but from the Asynchronous Advantage Option-Critic model presented in [5]. As in [5] we use an $\varepsilon$-soft policy leveraging the respective critic instead of learning a separate top level actor. As in [1] we potentially add in a regularization term $\eta$ for the termination policy update rule to decrease the likelihood that options terminate. In all of our experiments we used a discount factor of 0.99.

### 2.1 Exploring four rooms

**Hyperparameter search:** For the primitive actor-critic model our only tuned parameter is the learning rate over the range {0.001,0.01,0.1,0.25,1.0,10.0}. For the option-critic model we search over the number of options {4,8,16} and for the hierarchical option-critic model we use two options per layer of abstraction. All of our option models search over a intra-option learning rate shared among policies in the range {0.01,0.1,0.5}, a termination policy learning rate in the range {0.01,0.1,0.25,1.0} and a learning rate for critic models in the range {0.1,0.5}.

**Selected hyperparameters:** For actor-critic learning we found it best to use a learning rate of 0.01, and a temperature of 0.1. For option-critic and hierarchical option critic learning we found it optimal to use a temperature of 1.0, a learning rate of 0.5 for the critics and intra-options policies, and a learning rate of 0.25 for the termination policies. It was best to use 4 options for option-critic learning.

**Learning curve details:** We report the average number of steps taken in the last 100 episodes every 100 episodes, reporting the average of 50 runs with different random seeds for each algorithm.

### 2.2 Discrete stochastic decision process

**Hyperparameter search:** For the primitive actor-critic model our only tuned parameter is the learning rate over the range {0.001,0.01,0.1,0.25,1.0,10.0}. For the option-critic model we search over the number of options {4,8,16} and for the hierarchical option-critic model we use two options per layer of abstraction. All of our option models search over an intra-option learning rate shared among policies in the range {0.01,0.1,0.5}, a termination policy learning rate in the range {0.01,0.1,0.25,1.0} and a learning rate for critic models in the range {0.1,0.5}.

**Selected hyperparameters:** A learning rate of 0.25 is used for actor-critic learning and the critics of the option architectures have a learning rate of 0.5. We found it beneficial to use higher temperatures with higher levels of abstraction using 0.01 for one level, 0.1 for two levels and 1.0 for three levels. For the option-critic architecture we found it optimal to use an intra-option learning rate of 0.1, and a termination learning rate of 0.01. For the hierarchical option-critic architecture we found it optimal to use an intra-option learning rate of 1.0, and a termination learning rate of 10.0. 4 options was best for the option-critic model.

**Learning curve details:** We report the average reward over the last 100 episodes every 100 episodes, reporting the average of 10 runs with different random seeds for each algorithm.

### 2.3 Multistory building navigation

**Architecture details:** A core perceptual and contextualization model is shared across all policies and critics for each model to transform observations into conceptual states that can be processed

**Algorithm 1** Hierarchical Option-Critic with Tabular Intra-option Q-Learning

---

**procedure** LEARNEPISODE($env, N, \alpha, \gamma, \pi, \beta, \eta$)
    // get initial state
    $s \leftarrow s_0$
    // select options for initial state
    **for** $j = 1, ..., N-1$ **do**
        $o^j \leftarrow \pi^j(s, o^{1:j-1})$
    **repeat**
        // take an action and step through the environment
        $a \leftarrow \pi^N(a|s, o^{1:N-1})$
        $s', r \leftarrow env.step(a)$
        // calculate the expected discounted return
        $r' \leftarrow r$
        **if** $s'$ is non-terminal **then**
            $r' \leftarrow r' + \gamma U(s', o^{1:N-1})$ (see equation (10))
        // update the critic networks
        **for** $j = 1, ..., N-1$ **do**
            $\delta_j \leftarrow r' - Q_U(s, o^{1:j})$
            $Q_U(s, o^{1:j}) \leftarrow Q_U(s, o^{1:j}) + \alpha \delta_j$
        $\delta_N \leftarrow r' - Q_U(s, o^{1:N-1}, a)$
        $Q_U(s, o^{1:N-1}, a) \leftarrow Q_U(s, o^{1:N-1}, a) + \alpha \delta_N$
        // update the intra-option policies
        **for** $j = 1, ..., N-1$ **do**
            $\theta^j \leftarrow \theta^j + \alpha_\theta \frac{\partial log\pi^j(o^j|s, o^{1:j-1})}{\partial \theta^j} Q_U(s, o^{1:j})$
        $\theta^N \leftarrow \theta^N + \alpha_\theta \frac{\partial log\pi^N(a|s, o^{1:N-1})}{\partial \theta^N} Q_U(s, o^{1:N-1}, a)$
        // update the termination policies
        **for** $j = 1, ..., N-1$ **do**
            $\phi^j \leftarrow \phi^j - \alpha_\phi \prod_{i=j+1}^{N-1} \beta^i(s, o^{1:i}) \frac{\partial \beta^j(s, o^{1:j})}{\partial \phi^j} (A(s, o^{1:j}) + \eta)$
        // check which options have terminated and select new ones
        $o^{1:N-1} \leftarrow chooseTerminatedOptions(s', o^{1:N-1}, \pi, \beta, N)$
        // update the next state to now be the current state
        $s \leftarrow s'$
    **until** $s'$ is terminal
**procedure** CHOOSETERMINATEDOPTIONS($s, o^{1:k}, \pi, \beta, k$)
    **if** $\beta^k(s, o^{1:k}) = 1$
        **if** $k - 1 = 1$
            $o^1 \leftarrow \pi^1(s)$
        **else**
            $o^{1:k-1} \leftarrow chooseTerminatedOptions(s, o^{1:k-1}, \pi, \beta, k-1)$
        $o^k \leftarrow \pi^{k-1}(s, o^{1:k-1})$
    **return** $o^{1:k}$

---

---

**Algorithm 2** Asynchronous Advantage Hierarchical Option-Critic

---

**procedure** LEARNEPISODE($env, N, \alpha, \gamma, \pi, \beta, \eta, T_{max}, t_{min}, t_{max}$)
    initialize global counter $T \leftarrow 1$
    initialize thread counter $t \leftarrow 1$
    **repeat**
        $t_{start} = t$
        $s_t \leftarrow s_0$
        // reset gradients
        $dw \leftarrow 0$
        $d\theta \leftarrow 0$
        $d\phi \leftarrow 0$
        // select options for initial state
        **for** $j = 1, ..., N-1$ **do**
            $o_t^j \leftarrow \pi^j(s_t, o_t^{1:j-1})$
        **repeat**
            // take an action and step through the environment
            $a_t \leftarrow \pi^N(s_t, o_t^{1:N-1})$
            $s_{t+1}, r_t \leftarrow env.step(a_t)$
            // check which options have terminated and select new ones
            $o_t^{1:N-1} \leftarrow chooseTerminatedOptions(s_{t+1}, o_{t-1}^{1:N-1}, \pi, \beta, N)$
            $t \leftarrow t + 1$
            $T \leftarrow T + 1$
        **until** episode ends or $t - t_{start} == t_{max}$ or $(t - t_{start} > t_{min})$
        $G = V(s_t)$
        **for** $k = t - 1, ..., t_{start}$ **do**
            // accumulate thread specific gradients
            $G \leftarrow r_k + \gamma G$
            // update the critic policies
            **for** $j = 1, ..., N-1$ **do**
                $dw^j \leftarrow dw^j + \alpha_w \frac{\partial (G - Q(s, o^{1:j}))^2}{\partial w^j}$
            // update the intra-option policies
            **for** $j = 1, ..., N-1$ **do**
                $d\theta^j \leftarrow d\theta^j + \alpha_\theta \frac{\partial log \pi^j(o^j|s, o^{1:j-1})}{\partial \theta^j}(G - Q(s, o^{1:j-1}))$
                $d\theta^N \leftarrow d\theta^N + \alpha_\theta \frac{\partial log \pi^N(a|s, o^{1:N-1})}{\partial \theta^N}(G - Q(s, o^{1:N-1}, a))$
            // update the termination policies
            **for** $j = 1, ..., N-1$ **do**
                $d\phi^j \leftarrow d\phi^j - \alpha_\phi \prod_{i=j+1}^{N-1} \beta^i(s, o^{1:i}) \frac{\partial \beta^j(s, o^{1:j})}{\partial \phi^j}(A(s, o^{1:j}) + \eta)$
        update global parameters with thread gradients
    **until** $T > T_{max}$
**procedure** CHOOSETERMINATEDOPTIONS($s, o^{1:k}, \pi, \beta, k$)
    **if** $\beta^k(s, o^{1:k}) = 1$
        **if** $k - 1 = 1$
            $o^1 \leftarrow \pi^1(s)$
        **else**
            $o^{1:k-1} \leftarrow chooseTerminatedOptions(s, o^{1:k-1}, \pi, \beta, k-1)$
        $o^k \leftarrow \pi^{k-1}(s, o^{1:k-1})$
    **return** $o^{1:k}$

---

to produce an option policy. The perceptual module was a 100 unit fully connected layer with ReLU activations. This perceptual module is processed by a 256 unit LSTM network with gradients truncated at 20 steps. Every intra-option policy, termination policy, and critic simply consists of one linear layer on top of this core module followed by a softmax in the case of intra-option policies and a sigmoid in the case of termination policies.

**Hyperparameters:** We found optimal to use a learning rate of 1e-4 for all models a well as 16 parallel asynchronous threads and entropy regularization of 0.01 on the intra-option policies [1, 5]

**Learning curve details:** We set our implementation of A3C to report recent learning performance after approximately 1 minute of training. Each minute we report the rolling mean reward calculated using a horizon of 0.99. To plot learning performance we take the average and standard deviation of the reported rewards over the past 1 million frames.

## 2.4   Atari multi-task learning

**Experiment details:** In our Atari experiments we leverage the standard Open AI Gym v0 environments. A core perceptual and contextualization model is shared across all policies and critics for each model to transform observations into conceptual states that can be processed to produce primitive action and option policies. We follow architecture conventions for Atari games from [9] to implement this module consisting of a convolutional layer with 16 filters of size 8x8 with stride 4, followed by a convolutional layer with with 32 filters of size 4x4 with stride 2, followed by a fully connected layer with 256 hidden units. All three hidden layers were followed by a ReLU nonlinearity. This hidden representation is fed to a 256 unit LSTM network with gradients truncated at 20 steps. Every intra-option policy, termination policy, and critic simply consists of one linear layer on top of this core module followed by a softmax in the case of intra-option policies and a sigmoid in the case of termination policies. The primitive action policy for each game is implemented with its own linear layer followed by a softmax as the games have different action spaces. In our experiments on Atari we followed conventions from past work using 16 parallel asynchronous threads and entropy regularization of 0.01 on the intra-option policies [1, 5]. We use a learning rate of 1e-4 for each model.

**Analysis of learned options for multi-task learning:** In Table 1 we detail the average option switching frequencies for each of the 21 Atari games when we train in a many task learning setting. For the option-critic architecture and three-level hierarchical option-critic architecture we define a switch as terminating an option at a particular level and choosing a new different option at that level. We can see that the hierarchical option-critic architecture displays much greater variation in its option switching frequencies across games.

**Details on figures analyzing options:** In the main text we provide option specialization across Atari games for all 9 possible option combinations for the hierarchical option-critic architecture and the top 9 most used options for the option-critic architecture to save space. In Figure 1 we provide detailed information including the specialization of all learned options for the option-critic architecture. In all of our option analysis figures we use a heat-map where each option is assigned a color. This way options can be clearly separated from the surrounding options on the grid. We keep cells for options that are used on a game less than 1% of the time white. We then add a light color that gets progressively darker at 5% specialization, 10% specialization, and 25% specialization.

## 2.5   Comparison with methods for multi-task and lifelong learning

In this work we explore a relatively straightforward application of multi-task learning on the Atari games. Following conventions in multi-task learning [2], as the action space is different with varying sizes across games, all parts of the network are shared with the exception of a task specific layer in the last layer of the policy over primitive actions. This a somewhat arbitrary choice of the extent of weight sharing in light of recent work that focuses on more dynamic sharing patterns in multi-task learning, lifelong learning, and continual learning settings [10, 8, 13, 15, 3, 14, 6, 7, 11, 12]. A more dynamic weight sharing pattern should allow the hierarchical option-critic architecture to potentially achieve better sample efficiency in a multi-task learning setting. However, we leave analysis of the proper way to achieve this in a general sense to future work as it is largely orthogonal to our main contribution of presenting policy gradient theorems to optimize a deep hierarchy of options.

| Environment | OC | HOC ($o^1$) | HOC ($o^2$) |
|---|---|---|---|
| Alien | 5.4 | 7.7 | 1.5 |
| Amidar | 5.5 | 6.5 | 1.7 |
| Assault | 4.0 | 3.3 | 1.9 |
| Atlantis | 5.3 | 6.9 | 1.7 |
| BankHeist | 5.5 | 7.8 | 2.6 |
| BattleZone | 5.0 | 6.6 | 1.8 |
| BeamRider | 5.4 | 3.2 | 1.8 |
| Berzerk | 5.4 | 6.6 | 1.9 |
| Carnival | 5.5 | 4.4 | 2.0 |
| Centipede | 4.3 | 6.7 | 3.1 |
| ChopperCommand | 5.5 | 6.3 | 1.6 |
| DemonAttack | 5.4 | 3.4 | 1.7 |
| Jamesbond | 4.8 | 6.5 | 1.7 |
| MsPacman | 5.5 | 7.6 | 7.5 |
| Phoenix | 4.5 | 3.2 | 1.9 |
| Riverraid | 5.0 | 7.7 | 1.5 |
| Solaris | 3.4 | 5.6 | 2.7 |
| SpaceInvaders | 4.1 | 6.0 | 2.6 |
| Tutankham | 5.2 | 9.7 | 7.8 |
| WizardOfWor | 3.9 | 9.1 | 2.2 |
| Zaxxon | 5.5 | 4.2 | 1.7 |

Table 1: The average number of steps before switching options by game for the median performance option-critic (OC) and hierarchical option-critic (HOC) architectures during the evaluation period. For our three level model, we detail statistics for high level option $o^1$ as well as low level option $o^2$.

| Environment | o=0 | o=1 | o=2 | o=3 | o=4 | o=5 | o=6 | o=7 | o=8 | o=9 | o=10 | o=11 | o=12 | o=13 | o=14 | o=15 |
|---|---|---|---|---|---|---|---|---|---|---|---|---|---|---|---|---|
| Alien | 0.9% | 4.5% | 0.6% | 0.3% | 0.7% | 0.1% | 4.4% | 4.4% | 1.1% | 4.4% | 1.3% | 18.5% | 2.7% | 1.1% | 0.9% | 4.1% |
| Amidar | 0.8% | 4.5% | 0.5% | 0.3% | 0.5% | 0.1% | 4.5% | 4.4% | 1.1% | 4.4% | 1.3% | 18.6% | 2.7% | 1.1% | 0.9% | 4.1% |
| Assault | 0.9% | 5.0% | 5.0% | 0.3% | 1.5% | 7.3% | 5.3% | 7.2% | 1.4% | 10.9% | 1.3% | 0.1% | 5.7% | 56.9% | 1.0% | 4.6% |
| Atlantis | 77.0% | 4.5% | 0.4% | 0.3% | 0.3% | 0.1% | 4.4% | 5.1% | 1.1% | 4.3% | 72.5% | 0.1% | 2.7% | 1.1% | 0.9% | 5.7% |
| BankHeist | 0.9% | 4.6% | 0.6% | 0.3% | 0.5% | 0.1% | 4.4% | 4.3% | 1.1% | 4.3% | 1.3% | 18.6% | 2.7% | 1.1% | 0.9% | 4.0% |
| BattleZone | 0.9% | 4.5% | 0.4% | 37.3% | 1.1% | 0.1% | 4.5% | 4.4% | 1.1% | 4.3% | 1.3% | 0.1% | 10.7% | 1.2% | 7.4% | 4.0% |
| BeamRider | 0.9% | 5.5% | 1.4% | 0.4% | 0.3% | 13.6% | 4.5% | 4.4% | 1.1% | 4.4% | 1.3% | 0.1% | 2.7% | 1.1% | 0.9% | 4.0% |
| Berzerk | 1.0% | 4.5% | 0.6% | 0.3% | 0.3% | 13.6% | 4.4% | 7.8% | 1.1% | 4.3% | 2.2% | 0.1% | 2.7% | 1.1% | 0.9% | 4.0% |
| Carnival | 0.9% | 4.5% | 1.0% | 0.3% | 0.3% | 13.4% | 4.4% | 4.5% | 1.1% | 4.3% | 1.3% | 0.1% | 2.7% | 4.1% | 0.9% | 4.0% |
| Centipede | 1.2% | 5.6% | 2.7% | 3.3% | 0.6% | 10.3% | 6.3% | 5.1% | 2.0% | 6.8% | 1.5% | 0.2% | 4.1% | 16.3% | 2.3% | 7.9% |
| ChopperCommand | 0.9% | 4.5% | 0.5% | 0.3% | 0.5% | 0.1% | 4.5% | 4.4% | 1.1% | 4.3% | 1.3% | 18.6% | 2.7% | 1.1% | 0.9% | 4.0% |
| DemonAttack | 0.9% | 4.6% | 0.4% | 0.3% | 0.3% | 13.7% | 4.5% | 4.3% | 1.3% | 4.4% | 1.3% | 0.2% | 3.2% | 1.6% | 1.3% | 4.9% |
| Jamesbond | 0.9% | 4.8% | 0.4% | 0.3% | 0.3% | 13.2% | 8.3% | 4.7% | 4.4% | 4.3% | 1.3% | 0.1% | 2.7% | 1.1% | 0.9% | 12.1% |
| MsPacman | 0.9% | 4.5% | 0.5% | 0.3% | 49.1% | 0.1% | 4.4% | 4.3% | 1.1% | 4.3% | 1.3% | 0.1% | 2.7% | 1.1% | 0.9% | 4.0% |
| Phoenix | 0.9% | 4.5% | 49.2% | 1.4% | 3.6% | 0.1% | 4.4% | 4.4% | 1.1% | 4.3% | 1.3% | 1.1% | 2.7% | 1.1% | 0.9% | 4.0% |
| Riverraid | 0.9% | 4.4% | 0.7% | 0.3% | 5.1% | 0.1% | 4.5% | 4.4% | 1.1% | 4.3% | 1.3% | 16.8% | 2.7% | 1.1% | 0.9% | 4.1% |
| Solaris | 5.6% | 4.6% | 0.5% | 15.6% | 1.1% | 0.1% | 4.5% | 4.4% | 73.4% | 4.3% | 1.6% | 0.2% | 32.8% | 1.2% | 4.1% | 4.1% |
| SpaceInvaders | 1.3% | 6.6% | 27.3% | 10.4% | 0.3% | 0.2% | 4.4% | 4.4% | 1.2% | 4.4% | 1.3% | 0.1% | 2.7% | 2.9% | 33.7% | 4.2% |
| Tutankham | 1.0% | 4.5% | 2.7% | 27.5% | 0.4% | 0.1% | 4.5% | 4.3% | 1.1% | 4.4% | 1.3% | 0.2% | 2.7% | 1.1% | 37.5% | 4.0% |
| WizardOfWor | 0.9% | 4.6% | 3.2% | 0.3% | 32.3% | 0.1% | 4.5% | 4.3% | 1.1% | 4.3% | 1.3% | 5.6% | 2.7% | 1.2% | 0.9% | 4.0% |
| Zaxxon | 0.9% | 4.5% | 1.5% | 0.3% | 0.4% | 13.5% | 4.5% | 4.4% | 1.1% | 4.3% | 1.3% | 0.2% | 2.7% | 1.1% | 0.9% | 4.1% |

Figure 1: Option specialization across Atari games for a 16 option Option-Critic architecture trained in the many task learning setting.

Our approach is also orthogonal to recent approaches improving the efficiency of multi-task learning through a learned curriculum learning process [16, 4]. In the setting we explore, all models train on the games in a balanced fashion throughout time and the agent is not assumed to have any control over which environment it trains on. Controlling the curriculum of games to train on could also potentially improve the efficacy of our approach.