[Reviews · NeurIPS 2018]

Reviewer 1



The paper extends the Option-Critic policy gradient theorems to multiple levels of abstractions (i.e. it allows options-over options at multiple levels). The results they present apply to execution models where action choices are conditioned on a vector [o^1, o_2, ..., o_N] representing abstract action choices at levels 1,2,...,N (1 being the highest). These choices are determined using policies \pi (o^j | o_{j-1}, o_{j-2}, ..., 1). Moreover, the conditioning is maintained until a termination signal, modeled in a similar way, is triggered at some level j, at which point [o_j , o_{j+1}, ..., o_N] are updated. I consider this line of work very important, as it provides an investigation in the use of multiple levels of temporal abstractions, something that in most previous work was only alluded but never provided with clear derivations or empirical results. This paper provides these extensions. Although the paper is overall very well presented, there are some major technical details that are not properly described which makes me slightly skeptical of accepting the paper. See below for aspects that the authors should consider in improving the quality of the paper: Major issues: • From Eq (10), it seems that \beta(s, o^1, ..., o^N) is the prob. that at least one of the options terminate. For example, you use (1-\beta(s, o^1, ..., o^N)) for the event that none terminate. Still, on L170 you give a different definition: \beta(s, o^1, ..., o^{N-2}) is the prob. that o_{N-2} terminates given that o_{N-1} terminated and we were executing o^1, o_2, ..., o_{N-1}. This alternative definition is also used in the algorithm provided in the supplemental material. Although it makes no difference when N=2, it seems like a big inconsistency for N \geq 3. Please clarify the semantics of \beta to validate equation 10 and the Markov Chain in the supplemental material. • The paper argues for higher level abstractions, but nowhere in the experimental section do the authors try to illustrate the learned abstractions nor do they assess whether the resulting abstract options are degenerate (i.e. either never terminate or terminate immediately). • L268- L270: Can you please add the details on how you ensured a fair comparison between the 2-level and multi-level architectures? I think this is a crucial aspect which should not be left out in the supplemental material. E.g. what was the numerical limit to put on the number of options at each level? • as claimed in the abstract, the work "[learns abstract options] without the need for any intrinsic reward (IR) or subgoals", but it fails to explain why this is necessarily an advantage. I would argue that many of the problems associated with option-critic can be addressed by learning IR and subgoals (e.g. as demonstrated by work on Feudal Networks). The authors should clarify their perspective on the disadvantage of using IR and subgoals. • Minor: • nipick : what about "Hierarchical Option Critic" for a more concise title? • L52: benefits • L74: Can you give some thoughts on learning both latent goal representations and options achieving these subgoals? Te statement on L74 seems to imply that these are orthogonal. • L91: \pi: S \to (A \to [0,1]) for consistency with definition of P on L88. • L99: P(s_t=s | s_0, \pi_\theta) - state transitions are dependent on the policy • L110 - L129: You provide too much details coming from the Option-Critic paper - it would be much more useful to replace the policy gradient results (which are indirectly presented in 4.3) with a diagram that concisely illustrates the building blocks in Option-Critic as related to your work (e.g. the dependence between Q_\Omega, U, Q_U, V_\Omega, A_\Omega). • Section 4 + supplemental work: consider using o_{1:k} instead of "o^1, o^2, ..., o^k". All results would read much easier. • L185 (eq. 8) typo? Q_\Omega -> Q_U? • nipick: L351: remove "highly" + L352: I don't think the experimental results support the statement as used in the conclusion. The results are interesting and illustrate in some cases the benefits of multiple levels of abstraction, but I would be reserved in using such superlatives. ----After author feedback---- I would like to thank the authors for the additional feedback they provided, which includes clarifications to some of the issues I have raised. I still believe that the a proper qualitative analysis of the options learned at the higher levels is missing. I also find that a proper comparison to Feudal Networks is necessary for this work, and the author feedback is not very convincing on the many interesting aspects that Feudal Networks are using, such as intrinsic reward.

Reviewer 2



In this paper, the option-critic architecture is extended to a more general formulation of a multi-level hierarchy where instead of having just two levels (primitive actions and options), it is possible to have multiple option layers of different abstraction. Results on multiple benchmark tasks show that using N = 3 might be better than N = 2 for hierarchical RL learning. Pros: Studying and exploring a truly general hierarchical reinforcement learning architecture is of great importance. I share a similar view with the authors: although extending HRL to more than two layers seems to be a intuitive and desirable direction, there's not much attention in the field. The paper is well written and the technical contribution of developing option value function and subgoal policy gradient theorem seems solid. The flow of the paper is easy to follow. Cons: I'm not fully convinced that "the performance of the hierarchical option-critic architecture is impressive" (line 351). Compared to option-critic, the improvement of N=3 seems marginal. What's disappointing, but also important is the negative result when N = 4 in Rooms env. It basically says that, at least for the simple tasks we have right now, using more abstract options does not yield increasingly better performance. Besides the learning curves, there's no visualization of options learned in a higher level. Showing some option visualization on seaquest game might be helpful. There's no clear evidence that learning more abstract options is the way to go. I do think evaluating on harder games like Montezuma’s Revenge or continuous control tasks are necessary if we really want to know if HRL using multilevel abstract options works, though this might be beyond the scope of this paper. Another direction is to use grounded options with some human priors for specific tasks so we can examine if the options are learned correctly. The authors mentioned FuN which decomposes the state space could be an alternative. What's the advantage/disadvantage of building on top of FuN and learn multi-level subgoals, compared to hierarchical option-critic? Have you tried to use different temporal horizon for the higher level options in the hierarchy? The paper is very crowded and the figures are hard to read, hope the authors could find away to fix. a tiny thing I noticed: line 194 missing a period after "Appendix". Overall I think the paper is interesting and worth a larger audience despite the unsurprising experimental results.

Reviewer 3



I'm happy with authors rebuttal response. I'm improving the score under the assumption that the extended empirical analysis will be included in the final version ---------------- The paper proposes a generalization of Option-Critic (OC) to more than two levels. It derives respective policy gradient updates for termination and intra-option policy. The paper is quite clear, given the exhaustive supplementary material, can be well reproduced/followed both theoretically and empirically. I have found the experimental evaluation to be lacking, I hope that authors can elucidate the following: - What are the options that have actually been learnt on each of the levels? What is their duration? Are they non-trivial? From the presented experiments it is hard to see whether learnt options amount to anything significant or simply re-learn action-repeat heuristic or even collapse completely to micromanagement (option duration of 1). - Authors are right to point out the inconsistencies in ATARI evaluations. However, they make a very particular choice of using ATARI environment without action repeat (NoFrameskip). This is a non-standard setup and although there is nor reason not to use it, a discussion on why this choice was made is required. If proposed method only achieves improvement in this setup, this is fair enough, but it should be explicitly stated and discussed. Also, reporting results on a more standard (action repeat = 4) setup is then required. Going back to the first question, so what options does the method learn in the chosen (NoFrameskip) setup? Does it go beyond just learning the action repeat heuristic? What are the actual policies that it learns? - It is important to notice that the reported scores for the ATARI games are, overall, extremely low. For example, on the game where most improvement is observed (Zaxxon) scores reported in the literature ([30,1,18]) are on the order of tens of thousands, compared to 300-400 in this paper. This, surely, has to be discussed! The learning curve on Berzerk looks odd, is it contains no variance. What policy did it learn? I believe this discrepancy with the literature is due to not using action repeat, as discussed above, but it is unsatisfactory that authors don't address this explicitly in the manuscript at all. Given the comments above, I find the experimental section of the paper significantly below the standards of NIPS. I do appreciate the theoretical contribution of the paper and would've voted accept if the experimental section was done up to the standard. The experiments have to clearly answer: i) what options are actually learnt and are they meaningful? ii) do the help in a more standard setup? iii) how do they help in the setup chosen by the authors (did they re-learn action repeat?).